# Towards General Function Approximation in Zero-Sum Markov Games

**Baihe Huang**
School of Mathematical Sciences
Peking University
baihehuang@pku.edu.cn

**Jason D. Lee**
Department of Electrical and Computer Engineering
Princeton University
Jasondl@princeton.edu

**Zhaoran Wang**
Departments of Industrial Engineering & Management Sciences
Northwestern University
zhaoranwang@gmail.com

**Zhuoran Yang**
Department of Statistics and Data Science
Yale University
zhuoran.yang@yale.edu

## Abstract

This paper considers two-player zero-sum finite-horizon Markov games with simultaneous moves. The study focuses on the challenging settings where the value function or the model is parameterized by general function classes. Provably efficient algorithms for both decoupled and coordinated settings are developed. In the decoupled setting where the agent controls a single player and plays against an arbitrary opponent, we propose a new model-free algorithm. The sample complexity is governed by the Minimax Eluder dimension—a new dimension of the function class in Markov games. As a special case, this method improves the state-of-the-art algorithm by a $\sqrt{d}$ factor in the regret when the reward function and transition kernel are parameterized with $d$-dimensional linear features. In the coordinated setting where both players are controlled by the agent, we propose a model-based algorithm and a model-free algorithm. In the model-based algorithm, we prove that sample complexity can be bounded by a generalization of Witness rank to Markov games. The model-free algorithm enjoys a $\sqrt{K}$-regret upper bound where $K$ is the number of episodes.

## 1 Introduction

In competitive reinforcement learning, there are two agents competing against each other by taking actions. Their actions together determine the state evolutions and rewards. Function approximation, especially deep neural networks (LeCun et al., 2015), contributes to the success of RL in many real world applications, such as Atari (Mnih et al., 2013), Go (Silver et al., 2015), autonomous driving (Shalev-Shwartz et al., 2016), Texas holdem poker (Sandholm, 2017), and Dota (Berner et al., 2019).

The goal of competitive reinforcement learning aims to learn the Nash Equilibrium (NE) policy in a trial-and-error fashion. In a NE, no agent can be better off by unilaterally deviating from her policy. Most of the existing sample efficient competitive RL algorithms focus on the tabular or linear function approximation cases (Pérolat et al., 2017; Bai & Jin., 2020; Bai et al., 2021; 2020; Xie et al., 2020; Zhang et al., 2020). The huge empirical success of neural network based RL methods has remained largely unexplained. Thus, there exists a wide gap between theory and practice in competitive RL with general function approximation. One demanding question to ask is:

*Can we establish provably efficient competitive RL algorithms with general function approximation?*

In this paper, we make a step towards answering this question by providing structural conditions and complexity measures of Markov Games and function classes that allow for efficient learning. We focus on episodic zero-sum Markov Game (MG) with simultaneous move, where each episode consists of $H$ time steps and two players act simultaneously at each time step. The state space is arbitrarily large and can be infinite. We consider two function approximation settings: 1) use a general function class $F$ to approximate the action-value function (Q function); 2) use a general function class $\mathcal{M}$ to approximate the environment.

To this end, we introduce algorithms based on optimistic principles. Due to subtle game-theoretic issues, naive optimism is no longer working in Markov games where minimax optimization is performed. To deal with this issue, we use the algorithmic idea called 'alternate optimism'. Specifically, we develop algorithms for both coordinated and decoupled setting. In the decoupled setting, the agent controls one player and plays against an arbitrary and potentially adversarial opponent. The sample efficiency is measured by the gap between the learned value and value of the Nash Equilibrium. In the coordinated setting, the agent controls both players and the goal is to find the approximate Nash Equilibrium, i.e. with small duality gap. We identify key complexity measures of function classes to examine the effectiveness of elimination. By doing so, we prove upper bounds on sample complexity and regrets of the presented procedures that are independent of the number of states.

Our contributions are summarized into the following two folds:

- In the decoupled setting, we introduce Minimax Eluder dimension–a new complexity measure for competitive RL problems. We propose an algorithm that incurs at most $\widetilde{O}(H\sqrt{d_E K \log \mathcal{N}_{\mathcal{F}}})$ [1] regret in $K$ episodes where $d_E$ denotes the Minimax Eluder dimension $\mathcal{N}_{\mathcal{F}}$ denotes the covering number of function class, with probability at least $1 - p$. As a special case, this result improves Xie et al. (2020) by a $\sqrt{d}$ multiplicative factor in their setting when the reward function and transition kernel are linearly parameterized and $d$ is the dimension of feature mapping.

- In coordinated settings, we propose both model-based and model free algorithms. In the model-based setting, we generalize the witness rank (Sun et al., 2019a) to competitive RL. We prove that $\widetilde{O}(H^3 W^2/\epsilon^2)$ samples are enough to learn a policy $\epsilon$-close to the Nash Equilibrium, where $W$ is witness rank. In the model-free setting, we develop algorithm for agnostic learning with a candidate policy class $\Pi$. The algorithm incurs at most $\widetilde{O}(H\sqrt{dK \log(\mathcal{N}_{\mathcal{F}}\mathcal{N}_{\Pi})})$ regret. Here $d$ is a variant of Minimax Eluder dimension and $\mathcal{N}_{\Pi}$ denotes the covering number of policy class.

## 1.1 RELATED WORKS

There is a rich literature studying the learning and decision-making of Markov Games (Littman & Szepesvari, 1996; Greenwald et al., 2003; Grau-Moya et al., 2018; Pérolat et al., 2018; Srinivasan et al., 2018; Sidford et al., 2020; Wei et al., 2017; Pérolat et al., 2017; Bai & Jin., 2020; Bai et al., 2021; 2020; Zhang et al., 2020; Zhao et al., 2021). The most related to us are perhaps (Xie et al., 2020; Chen et al., 2021; Jin et al., 2021b), where the authors address the challenge of exploration-exploitation tradeoff in large state spaces. Due to space constraint, a detailed literature discussion is deferred to Appendix A.

## 2 PRELIMINARIES

We consider two-player zero-sum simultaneous-moves episodic Markov game, defined by the tuple $(\mathcal{S}, \mathcal{A}_1, \mathcal{A}_2, r, \mathbb{P}, H)$, where $\mathcal{S}$ is the state space, $\mathcal{A}_i$ is a finite set of actions that player $i \in \{1, 2\}$ can take, $r$ is the reward function, $\mathbb{P}$ is the transition kernel and $H$ is the number of time steps. At each time step $h \in [H]$, player P1 and P2 take actions $a \in \mathcal{A}_1$ and $b \in \mathcal{A}_2$ respectively upon observing the state $x \in \mathcal{S}$, and then both receive the reward $r_h(x, a, b)$. The system then transitions to a new state $x' \sim \mathbb{P}_h(\cdot|x, a, b)$ according to the transition kernel $\mathbb{P}$. Throughout this paper, we assume for simplicity that $\mathcal{A}_1 = \mathcal{A}_2 = \mathcal{A}$ and that the rewards $r_h(x, a, b)$ are deterministic functions of the tuple $(x, a, b)$ taking values in $[-1, 1]$. Turn-based games are special cases of simultaneous games in the

---

[1] We use $\widetilde{O}$ to hide logarithmic terms in $H$, $1/p$ and $K$.

sense that at each state the reward and transition are independent of one player's action (Xie et al., 2020). Generalizations to $\mathcal{A}_1 \neq \mathcal{A}_2$ and stochastic reward are also straightforward.

Denote by $\Delta := \Delta(\mathcal{A})$ the probability simplex over the action space $\mathcal{A}$. A stochastic policy of P1 is a length-$H$ sequence of functions $\pi := \{\pi_h : \mathcal{S} \mapsto \Delta\}_{h \in [H]}$. At each step $h \in [H]$ and state $x \in \mathcal{S}$, P1 takes an action sampled from the distribution $\pi_h(x)$ over $\mathcal{A}$. Similarly, a stochastic policy of P2 is given by the sequence $\nu := \{\nu_h : \mathcal{S} \mapsto \Delta\}_{h \in [H]}$. The learning happens in $K$ episodes. In each episode $k$, each player P1 and P2 proposes a policy $\pi^k$ and $\nu^k$ respectively based on history up to the end of episode $k-1$ and then executes the policy to observe the trajectories $\{x_h^k, a_h^k, b_h^k\}_{h=1}^H$.

For a fixed policy pair $(\pi, \nu)$, the value function and Q functions for zero-sum game is defined by

$$V_{h_0}^{\pi,\nu}(x) := \mathbb{E}\left[\sum_{h=h_0}^H r_h(x_h, a_h, b_h)|x_{h_0} = x\right],$$

$$Q_{h_0}^{\pi,\nu}(x,a,b) := \mathbb{E}\left[\sum_{h=h_0}^H r_h(x_h, a_h, b_h)|x_{h_0} = x, a_{h_0} = a, b_{h_0} = b\right],$$

where the expectation is over $a_h \sim \pi_h(x_h)$, $b_h \sim \nu_h(x_h)$ and $x_{h+1} \sim \mathbb{P}_h(\cdot|x_h, a_h, b_h)$. $V_1^{\pi,\nu}$ and $Q_1^{\pi,\nu}$ are often abbreviated to $V^{\pi,\nu}$ and $Q^{\pi,\nu}$. In zero-sum games, PI aims to maximize $V^{\pi,\nu}(x_1)$ and P2 aims to minimize $V^{\pi,\nu}(x_1)$. Given policy $\pi$ of P1, the best response policy of P2 is defined by $\nu_\pi^* := \arg\min_\nu V^{\pi,\nu}(x_1)$. Similarly given policy $\nu$ of P2, the best response policy of P1 is defined by $\pi_\nu^* := \arg\max_\pi V^{\pi,\nu}(x_1)$. Then from definitions,

$$V^{\pi,\nu_\pi^*}(x_1) \leq V^{\pi^*,\nu^*}(x_1) \leq V^{\pi_\nu^*,\nu}(x_1)$$

and the equality holds for the Nash Equilibrium (NE) of the game $(\pi^*, \nu^*)$. We abbreviate $V^{\pi^*,\nu^*}(x_1)$ as $V^*$ and $Q^{\pi^*,\nu^*}(x_1)$ as $Q^*$. $V^*$ is often referred to as the value of the game. As common in Markov games literature (Pérolat et al., 2015), we will use the following Bellman operator

$$\mathcal{T}_h Q_{h+1}(x,a,b) := r(x,a,b) + \mathbb{E}_{x' \sim \mathbb{P}_h(\cdot|x,a,b)}[\max_{\pi'} \min_{\nu'} \mathbb{E}_{\pi',\nu'} Q_{h+1}(x', \cdot, \cdot)], \qquad (1)$$

and the following Bellman operator for fixed P1's policy

$$\mathcal{T}_h^\pi Q_{h+1}(x,a,b) := r(x,a,b) + \mathbb{E}_{x' \sim \mathbb{P}_h(\cdot|x,a,b)}[\min_{\nu'} \mathbb{E}_{\pi,\nu'} Q_{h+1}(x', \cdot, \cdot)]. \qquad (2)$$

**Notations** For a integer $H$, $[H]$ denotes the set $\{1, 2, \ldots, H\}$. For a finite set $\mathcal{S}$, $|\mathcal{S}|$ denotes its cardinality. For a matrix $A \in \mathbb{R}^d$, $A_{i,*}$ and $A_{*,j}$ denote the $i$-th row and $j$-th column of $A$ respectively. For a function $f : \mathcal{S} \mapsto \mathbb{R}$, $\|f\|_\infty$ denotes $\sup_{s \in \mathcal{S}} |f(s)|$. We use $\mathcal{N}(\mathcal{F}, \epsilon)$ to denote the $\epsilon$-covering number of $\mathcal{F}$ under metric $d(f,g) = \max_h \|f_h - g_h\|_\infty$.

## 3 DECOUPLED SETTING

In decoupled setting, the algorithm controls P1 to play against P2. In $k$-th episode, P1 chooses a policy $\pi^k$ and P2 chooses a policy $\nu^k$ based on $\pi^k$. Then P1 and P2 interact in the Markov Game by playing $\pi^k$ and $\nu^k$ respectively, and observe the rewards and states in this episode. Notice that P2 can be adversarial and its policy is never revealed, meaning that P1 can only maximize its reward by playing $\pi^*$. The goal is to thus minimize the following 'value regret'

$$\text{Reg}(K) = \sum_{k=1}^K [V_1^*(x_1) - V_1^{\pi^k,\nu^k}(x_1)]. \qquad (3)$$

A small value regret indicates that we learn the value of MG. If an algorithm obtains an upper bound on value regret that scales sublinearly with $K$ for all $\nu^k$, then it can not be exploited by opponents.

### 3.1 FUNCTION APPROXIMATION

We consider function class $\mathcal{F} = \mathcal{F}_1 \times \cdots \times \mathcal{F}_H$ where $\mathcal{F}_h \subset \{f : \mathcal{S} \times \mathcal{A}_1 \times \mathcal{A}_2 \mapsto [0,1]\}$ to provide function approximations for $Q^*$ — the Q-function of Nash Equilibrium. For a policy pair $(\pi, \nu)$ and function $f$ we abuse the notations and use $f(x, \pi, \nu) = \mathbb{E}_{a \sim \pi, b \sim \nu}[f(x,a,b)]$ to denote the expected reward of executing $(\pi, \nu)$ in function $f$. We make the following assumptions about $\mathcal{F}$.

**Assumption 3.1** (Realizability). *We assume $Q_h^* \in \mathcal{F}_h$ for all $h \in [H]$.*

**Assumption 3.2** (Completeness). *We assume for all $h \in [H]$ and $f \in \mathcal{F}_{h+1}$, $\mathcal{T}_h f \in \mathcal{F}_h$.*

Realizability means the function class contains the target Q function. Completeness says the function class is closed under the Bellman operator. These assumptions are common in value function approximation literature (e.g. Jin et al. (2020); Wang et al. (2020b); Jin et al. (2021a)). Note that they are weaker than Xie et al. (2020), as their setting satisfies $\mathcal{T}_h f \in \mathcal{F}_h$ for any action-value function $f$.

For a function $f \in \mathcal{F}$, we denote the max-min policy of P1 by $\pi_f$. Specifically,

$$(\pi_f)_h(x) := \operatorname{argmax}_\pi \min_\nu f_h(x, \pi, \nu), \forall h \in [H].$$

Similarly we denote the min-max policy of P2 by $\nu_f$. Given policy $\pi$ of P1, the best response policy of P2 in terms of function $f$ is defined by

$$(\nu_\pi^f)_h(x) := \operatorname{argmin}_\nu f_h(x, \pi_h, \nu), \forall h \in [H].$$

Similarly given policy $\nu$ of P2, the best response policy of P1 in terms of function $f$ is denoted by $\pi_\nu^f$. When there is no confusion, we will drop the subscript $h$ for simplicity.

Learnability requires the function class to have bounded complexity. We introduce Minimax Eluder dimension to capture the structural complexity of function classes in zero-sum games. To define, we use $\epsilon$-independence of distributions and Distributional Eluder dimension (DE) from Russo & Van Roy (2013); Jin et al. (2021a).

**Definition 3.3** ($\epsilon$-independence of distributions). *Let $\mathcal{G}$ be a function class defined on $\mathcal{X}$, and $\nu, \mu_1, \ldots, \mu_n$ be probability measures over $\mathcal{X}$. We say that $\nu$ is $\epsilon$-independent of $\{\mu_1, \ldots, \mu_n\}$ with respect to $\mathcal{G}$ if there exists $g \in \mathcal{G}$ such that $\sqrt{\sum_{i=1}^n (\mathbb{E}_{\mu_i}[g])^2} \le \epsilon$ but $|\mathbb{E}_\nu[g]| \ge \epsilon$.*

**Definition 3.4** (Distributional Eluder dimension (DE)). *Let $\mathcal{G}$ be a function class defined on $\mathcal{X}$, and $\mathcal{D}$ be a family of probability measures over $\mathcal{X}$. The distributional Eluder dimension $\dim_{\mathrm{DE}}(\mathcal{G}, \mathcal{D}, \epsilon)$ is the length of the longest sequence $\{\rho_1, \ldots, \rho_n\} \subset \mathcal{D}$ such that there exists $\epsilon' \ge \epsilon$ where $\rho_i$ is $\epsilon'$-independent of $\{\rho_1, \ldots, \rho_{i-1}\}$ for all $i \in [n]$.*

Now we introduce Minimax Eluder dimension. Recall the minimax Bellman operator from Eq (1):

$$\mathcal{T}_h f_{h+1}(x, a, b) := r(x, a, b) + \mathop{\mathbb{E}}_{x' \sim \mathbb{P}_h(\cdot|x,a,b)}[\max_{\pi'} \min_{\nu'} f_{h+1}(x', \pi', \nu')]$$

thus Minimax Eluder dimension is the Eluder dimension on the Bellman residues with respect to the above minimax Bellman operator.

**Definition 3.5** (Minimax Eluder dimension). *Let $(I - \mathcal{T}_h)\mathcal{F} := \{f_h - \mathcal{T}_h f_{h+1} : f \in \mathcal{F}\}$ be the class of minimax residuals induced by function class $\mathcal{F}$ at level $h$ and $\mathcal{D}_\Delta = \{\mathcal{D}_{\Delta,h}\}_{h \in [H]}$ where $\mathcal{D}_{\Delta,h} := \{\delta(x, a, b) : x \in \mathcal{S}, a \in \mathcal{A}_1, b \in \mathcal{A}_2\}$ is the set of Dirac measures on state-action pair $(x, a, b)$. The $\epsilon$-Minimax Eluder dimension of $\mathcal{F}$ is defined by $\dim_{\mathrm{ME}}(\mathcal{F}, \epsilon) := \max_{h \in [H]} \dim_{\mathrm{DE}}((I - \mathcal{T}_h)\mathcal{F}, \mathcal{D}_\Delta, \epsilon)$.*

## 3.2 OPTIMISTIC NASH ELIMINATION FOR MARKOV GAMES

Now we introduce Optimistic Nash Elimination for Markov Games (ONEMG), presented in Algorithm 1. This algorithm maintains a confidence set $\mathcal{V}^k$ that always contains the $Q^*$, and sequentially eliminates inaccurate hypothesis from it. In $k$-th episode, P1 first optimally chooses a value function $f^k$ in $\mathcal{V}^{k-1}$ that maximizes the value of the game. Intuitively, this step performs optimistic planning in the pessimistic scenarios, i.e. assuming P2 plays the best response. Next, it plays against P2 and augments the data from this episode into replay buffer $\mathcal{B}$. The confidence set $\mathcal{V}^k$ is then updated by keeping the functions $f$ with small minimax Bellman errors $f_h - \mathcal{T}_h f_{h+1}$, in Line 12. To estimate $f_h - \mathcal{T}_h f_{h+1}$, we use $\mathcal{E}_{\mathcal{B}_h}(f_h, f_{h+1}) - \inf_{g \in \mathcal{F}_h} \mathcal{E}_{\mathcal{B}_h}(g, f_{h+1})$, a standard variance reduction trick to avoid the double-sample issue. It can be shown that when the value function class is complete, $\mathcal{E}_{\mathcal{B}_h}(f_h, f_{h+1}) - \inf_{g \in \mathcal{F}_h} \mathcal{E}_{\mathcal{B}_h}(g, f_{h+1})$ is an unbiased estimator of $f_h - \mathcal{T}_h f_{h+1}$ (Lemma B.1).

Notice that unlike many previous works that add optimistic bonus on every state-action pairs (Xie et al., 2020; Chen et al., 2021), Algorithm 1 only takes optimism in the initial state. Instead, the constraint set contains every function that has low Bellman residue on all trajectories in the replay buffer. This 'global optimism' technique trades computational efficiency for better sample complexity, and can be found in many previous work in Markov Decision process (Zanette et al., 2020; Jin et al., 2021a). As we will see later, it is also useful in the coordinated setting.

---

**Algorithm 1** Optimistic Nash Elimination for Markov Games (ONEMG)

---

1: **Input:** Function class $\mathcal{F}$
2: Initialize $\mathcal{B}_h = \emptyset, h \in [H], \mathcal{V}^0 \leftarrow \mathcal{F}$
3: Set $\beta \leftarrow C \log(\mathcal{N}(\mathcal{F}, 1/K) \cdot HK/p)$ for some large constant $C$
4: **for** $k = 1, 2, \ldots, K$ **do**
5:     Compute $f^k \leftarrow \mathrm{argmax}_{f \in \mathcal{V}^{k-1}} f_1(x_1, \pi_f, \nu_f)$, let $\pi_k \leftarrow \pi_{f^k}$
6:     **for** step $h = 1, 2, \ldots, H$ **do**
7:         Interact with environment by playing action $a_h^k \sim \pi_h^k(x_h^k)$
8:         Observe reward $r_h^k$, opponent's action $b_h^k$ and move to next state $x_{h+1}^k$
9:     **end for**
10:    Update $\mathcal{B}_h \leftarrow \mathcal{B}_h \cup \{(x_h^k, a_h^k, b_h^k, r_h^k, x_{h+1}^k)\}, \forall h \in [H]$
11:    For all $\xi, \zeta \in \mathcal{F}$ and $h \in [H]$, let

$$\mathcal{E}_{\mathcal{B}_h}(\xi_h, \zeta_{h+1}) = \sum_{\tau=1}^{k} [\xi_h(x_h^\tau, a_h^\tau, b_h^\tau) - r_h^\tau - \zeta_{h+1}(x_{h+1}^\tau, \pi_{\zeta_{h+1}}, \nu_{\zeta_{h+1}})]^2$$

12:    Set
$$\mathcal{V}^k \leftarrow \{f \in \mathcal{F} : \mathcal{E}_{\mathcal{B}_h}(f_h, f_{h+1}) \leq \inf_{g \in \mathcal{F}_h} \mathcal{E}_{\mathcal{B}_h}(g, f_{h+1}) + \beta, \forall h \in [H]\}$$

13: **end for**

---

### 3.3 THEORETICAL RESULTS

In this section we present the theoretical guarantee for Algorithm 1. The proof is deferred to Appendix B.

**Theorem 3.6.** *Under Assumption 3.1 and Assumption 3.2, the regret Eq* (3) *of Algorithm 1 is upper bounded by*

$$\mathrm{Reg}(K) \leq O\left(H\sqrt{K \cdot d_{\mathrm{ME}} \cdot \log(HK\zeta)}\right)$$

*with probability at least* $1 - p$. *Here* $d_{\mathrm{ME}} = \dim_{\mathrm{ME}}(\mathcal{F}, \sqrt{1/K})$ *and* $\zeta = \mathcal{N}(\mathcal{F}, 1/K)/p$.

As this theorem suggests, as long as the function class has finite Minimax Eluder dimension and covering number, P1 can achieve $\sqrt{K}$-regret against P2. Notice that when P2 always plays the best response of P1's policies, this regret guarantee indicates that algorithm eventually learns the Nash Equilibrium. In this case, the algorithm takes $\widetilde{O}(H^2 d \log(|\mathcal{F}|)/\epsilon^2)$ samples to learn a policy $\pi$ that is $\epsilon$-close to $\pi^*$.

When the opponent plays dummy actions, the Markov Game can be seen as an MDP and the value of the Markov Game is the maximum cumulative sum of rewards one can achieve in this MDP. In this case, our result reduces to $\widetilde{O}(H \cdot \sqrt{K \cdot d_{\mathrm{BE}} \log(\mathcal{N}(\mathcal{F}, 1/K))})$ where $d_{\mathrm{BE}}$ is the Bellman eluder dimension in Jin et al. (2021a).

In particular, when the function class is finite, the regret becomes $O(H\sqrt{Kd_{\mathrm{ME}} \log(HK|\mathcal{F}|/p)})$ that depends logarithmically on the cardinality of $\mathcal{F}$. Moreover, when the reward and transition kernel have linear structures, i.e. $r_h(x, a, b) = \phi(x, a, b)^\top \theta_h$ and $\mathbb{P}_h(\cdot|x, a, b) = \phi(x, a, b)^\top \mu_h(\cdot)$ where $\phi(x, a, b) \in \mathbb{R}^d$, then $d_{\mathrm{ME}}(\mathcal{F}, \sqrt{1/K}) = \log \mathcal{N}(\mathcal{F}, 1/K) = \widetilde{O}(d)$ and the regret becomes $\widetilde{O}(Hd\sqrt{K})$ [2]. Thus we improve Xie et al. (2020) by a $\sqrt{d}$ factor. We also provide a simpler algorithm tailored for this setting, see Appendix B.1 for details.

Algorithm 1 solves a non-convex optimization problem in Line 5 with highly non-convex constraint set (Line 12). In general, it is computationally inefficient. Even when reduced to linear MDP, i.e. the opponent plays dummy actions and the transition matrices and the rewards have low rank structures, it is not known if the same $\widetilde{O}(H\sqrt{d^2 K})$ regret can be achieved with computationally efficient algorithms (Zanette et al., 2020). Designing computationally efficient algorithms with near-optimal sample efficiency is an interesting further direction.

---

[2] Here we use $\widetilde{O}$ to omit the logarithm factors in $K$, $d$, $H$ and $1/p$.

## 4 COORDINATED SETTING

In the coordinated setting, the agent can control both P1 and P2. The goal is to find the $\epsilon$-approximate Nash equilibrium $(\pi, \nu)$ in the sense that

$$V^{\pi_\nu^*, \nu}(x_1) - V^{\pi, \nu_\pi^*}(x_1) \leq \epsilon \tag{4}$$

by playing P1 and P2 against each other in the Markov game. In the following sections, we propose both model-based and model-free methods to deal with this problem.

### 4.1 MODEL-BASED ALGORITHM

In model-based methods, we make use of a model class $\mathcal{M}$ of candidate models to approximate the true model $M^*$. We are also given a test function class $\mathcal{G}$ to track model misfit. Given a model $M \in \mathcal{M}$, we use $Q^M(x, a, b)$, $r^M(x, a, b)$ and $\mathbb{P}^M(\cdot|x, a, b)$ to denote the Q-function, reward function and transition kernel of executing action $a$ and $b$ at state $x$ in model $M$. For simplicity, we use $(r, x) \sim M$ to represent $r \sim r^M(x, a, b)$ and $x \sim \mathbb{P}^M(\cdot|x, a, b)$. We denote the NE policy in model $M$ as $\pi^M$ and $\nu^M$. For policy $\pi, \nu$ and model $M$, we use $\nu_\pi^M$ to denote the best response of $\pi$ in model $M$ and $\pi_\nu^M$ is defined similarly.

#### 4.1.1 ALTERNATE OPTIMISTIC MODEL ELIMINATION (AOME)

The model-based method, Alternate Optimistic Model Elimination (AOME), is presented in Algorithm 4. It maintains a constraint set $\mathcal{M}^k$ of candidate models. Throughout $K$ iterations, AOME makes sure that the true model $M^*$ always belongs $\mathcal{M}^k$ for all $k \in [K]$, and sequentially eliminates incorrect models from $\mathcal{M}^k$.

However, the idea of being optimistic only at initial state (Zanette et al., 2020; Jin et al., 2021a) is not directly applicable to this scenario. Since the objective to be optimized is the duality gap, the target policies being considered are thus policies pairs $(\pi_\nu^*, \nu)$ and $(\pi, \nu_\pi^*)$. However, $\pi_\nu^*$ and $\nu_\pi^*$ are not available to the agents. Therefore, the performances of proposed policies are evaluated on out-of-distribution trajectories, meaning that optimism can not be guaranteed. This causes the distribution shift issue. In worst cases, low Bellman errors in the collected trajectories provide no information for the trajectories collected by $\pi_\nu^*, \nu$ and $\pi, \nu_\pi^*$.

To address this issue, the algorithm performs global planning by applying the idea of *alternate optimism* to model-based methods. In $k$-th episode, the algorithm first solves the optimization problem $M_1^k = \arg\max_{M \in \mathcal{M}} Q^M(x_1, \pi^M, \nu^M)$ and let $\pi^k = \pi^{M_1^k}$. This optimization problem corresponds to finding the optimistic estimate of the value of the game. Indeed, notice that $M^* \in \mathcal{M}^k$ implies $Q^{M_1^k}(x_1, \pi^k, \nu^k) \geq Q^{M_1^k}(x_1, \pi^k, \nu_{\pi^k}^{M_1^k}) \geq V^*(x_1)$, by optimality of $M_1^k$. Next, it solves the second optimization problem $M_2^k = \arg\min_{M \in \mathcal{M}} Q^M(x_1, \pi^{M_1^k}, \nu^M)$ and let $\nu^k \leftarrow \nu_{\pi^k}^{M_2^k}$. This corresponds to finding the pessimistic estimate of $V^{\pi^k, \nu_{\pi^k}^*}(x_1)$. In fact, we see that $M^* \in \mathcal{M}^k$ implies $Q^{M_2^k}(x_1, \pi^k, \nu^k) \leq V^{\pi^k, \nu_{\pi^k}^*}(x_1)$, by optimality of $M_2^k$. This approach appears in Wei et al. (2017) to guarantee convergence to Nash equilibrium, where it is referred to as 'Maximin-EVI'.

In Line 8, the algorithm checks if the values of model $M_1^k$ and $M_2^k$ are close to the value of the true model $M^*$ when executing policy $\pi^k$ and $\nu^k$. If the condition holds, we have

$$V^*(x_1) - V^{\pi^k, \nu_{\pi^k}^*}(x_1) \leq Q^{M_1^k}(x_1, \pi^k, \nu^k) - Q^{M_2^k}(x_1, \pi^k, \nu^k) \leq \epsilon,$$

which means that AOME finds policy $\pi^k$ that is $\epsilon$-close the NE. One can also switch the order of alternate optimism and obtain $\nu^k$ that is $\epsilon$-close the NE. We then terminate the process and output $\pi^k$ and $\nu^k$. If the algorithm does not terminate in Line 8, it applies the witnessed model misfit checking method in Sun et al. (2019a). It starts by computing the empirical Bellman error of Markov games defined as follows

$$\widehat{\mathcal{L}}(M_1, M_2, M, h) := \sum_{i=1}^{n_1} \frac{1}{n_1} [Q_h^M(x_h^i, a_h^i, b_h^i) - (r_h^i + Q_h^M(x_{h+1}^i, \pi^{M_1}, \nu_{\pi^{M_1}}^{M_2}))]. \tag{5}$$

As proved in Appendix D, in this case $\widehat{\mathcal{L}}(M_1^k, M_2^k, M_j^k, h^k)$ must be greater than $\epsilon/(8H)$ for some $j \in [2]$ and $h^k \in [H]$. Then the algorithm samples additional trajectories at level $h^k$ and shrinks the constraint set by eliminating models with large empirical model misfit, which is defined as follow

$$\widehat{\mathcal{E}}(M_1^k, M_2^k, M, h^k) = \sup_{g \in \mathcal{G}} \sum_{i=1}^n \frac{1}{n} \mathop{\mathbb{E}}_{(r_h, x_h) \sim M} [g(x_h^i, a_h^i, b_h^i, r_h, x_h) - g(x_h^i, a_h^i, b_h^i, r_h^i, x_{h+1}^i)]. \quad (6)$$

We will show that this constraint set maintains more and more accurate models of the environment. Due to space constraints, the algorithm and theory of our model-based method are deferred to Appendix C.

## 4.2 Model free algorithm

In this section we propose a model-free algorithm. We consider a more general agnostic learning, where The algorithm is given a policy class $\Pi = \Pi_1 \times \cdots \Pi_H$ with $\Pi_h \subset \{\pi_h : \mathcal{S} \mapsto \Delta\}, h \in [H]$. Notice by letting $\Pi$ to be the class of NE policies induced by the action-value function class $\mathcal{F}$, we recover the original setup. The goal is to find policy $\pi$ and $\nu$ from $\Pi$ to minimize the duality gap: $\max_{\pi' \in \Pi} V^{\pi', \nu} - \min_{\nu' \in \Pi} V^{\pi, \nu'}$, by playing only policies from $\Pi$. Since only policies from $\Pi$ are considered, we overload the notation and define the optimal solution in $\Pi$ as $\pi^* = \arg\max_{\pi \in \Pi} \min_{\nu' \in \Pi} V^{\pi, \nu'}$ and $\nu^* = \arg\min_{\nu \in \Pi} \max_{\pi' \in \Pi} V^{\pi', \nu}$. When $\Pi$ contains all possible policies, $\pi^*$ and $\nu^*$ is then the Nash equilibrium.

Similar to Section 3, we consider general function class $\mathcal{F} = \mathcal{F}_1 \times \cdots \times \mathcal{F}_H$ where $\mathcal{F}_h \subset \{f : \mathcal{S} \times \mathcal{A}_1 \times \mathcal{A}_2 \mapsto [0,1]\}$. We overload the notations in Section 3 and use $\nu_\pi^f$ to denote the best response of $\pi$ from $\Pi$, namely $\nu_\pi^f(x) := \arg\min_{\nu \in \Pi} f(x, \pi, \nu)$. Similarly, we use $\pi_f(x) := \arg\max_{\pi \in \Pi} \min_{\nu \in \Pi} f(x, \pi, \nu)$. We also use $\nu_\pi^*(x) := \arg\min_{\nu \in \Pi} V^{\pi, \nu}$ to denote the best response of $\pi$ in the true model.

### 4.2.1 Alternate Optimistic Value Elimination (AOVE)

Notice that the duality gap can be decomposed as follows

$$\text{gap} = \underbrace{V^{\pi_\nu^*, \nu} - V^{\pi_{\nu^*}^*, \nu^*}}_{\text{P2}} + \underbrace{V^{\pi_{\nu^*}^*, \nu^*} - V^{\pi^*, \nu_{\pi^*}^*}}_{\text{Optimal}} + \underbrace{V^{\pi^*, \nu_{\pi^*}^*} - V^{\pi, \nu_\pi^*}}_{\text{P1}}$$

where the optimal part $V^{\pi_{\nu^*}^*, \nu^*} - V^{\pi^*, \nu_{\pi^*}^*}$ is fixed. Therefore we can consider only the P1 part here and the P2 follows by symmetry. We are interested in the following policy regret

$$\text{Reg}(K) := \sum_{k=1}^K (V^{\pi^*, \nu_{\pi^*}^*} - V^{\pi^k, \nu_{\pi^k}^*}). \quad (7)$$

Policy regret measures the extent that P1 can be exploited by policies in $\Pi$. Our algorithm Alternate Optimistic Value Elimination (AOVE), presented in Algorithm 2, works on minimizing this regret. Similar to Algorithm 1, it also uses optimism in the initial state to address exploration and exploitation trade-off and constructs a series of confidence sets of hypotheses with small Bellman errors. The differences primarily lie in a different choice of optimism and a different series of confidence sets constructed with the Bellman operator in Eq 2. Since the policy class $\Pi$ is taken into consideration, the algorithm maintains a constraint set $\mathcal{B} \subset \Pi \times \mathcal{F}$ of policy-function pairs. We use the Bellman operator with regard to fixed P1's policy, in Eq (2). Thus intuitively, this constraint set maintains $(\pi, f)$ pairs such that $f_h - \mathcal{T}_h^\pi f_{h+1}$ is small for all $h \in [H]$.

We apply the 'alternate optimistic' principle, presented in Line 5 and Line 6. Intuitively, the algorithm finds an optimistic planning of $\pi$ and pessimistic planning of $\nu$, together corresponding to a max-min procedure. As such, we form an upper bound $f_1(x_1, \pi, \nu_\pi^f) - g_1(x_1, \pi^k, \nu_{\pi^k}^g)$ of the duality gap. Note that it is different from the 'alternate optimistic' used in model-based settings Wei et al. (2017) in that the hypothesis chosen in Line 6 is constrained by the policy chosen in Line 5. This is because in model-based methods, 'plausible' models in the confidence set guarantee small simulation errors (Lemma C.6). However, this is not true for 'plausible' value functions in the confidence set.

The rest of the algorithm aims at minimizing this upper bound sequentially by eliminating bad hypothesis in the constraint set. The elimination process occurs in Line 13, where the algorithm uses history data to eliminate functions with large Bellman error. To estimate Bellman

error, we use the similar method as in Algorithm 1, i.e. subtracting the variance of $\mathcal{T}_h^\pi f_{h+1}$ by $g = \arg\inf_{g\in\mathcal{F}_h} \mathcal{E}_{\mathcal{D}_h}(g, f_{h+1}, \pi)$. Notice that a smaller value of $f_h - \mathcal{T}_h^\pi f_{h+1}$ indicates that function approximation $f$ is not easily exploited by P2 when P1 plays $\pi$.

As shown in Appendix D, $(\pi^*, Q^*)$ is always in the constraint set, and the Bellman errors of the rest hypothesis keep shrinking. Thus in the presented algorithm of AOVE, the regret of P1 part $V^{\pi^*, \nu_{\pi^*}^*} - V^{\pi, \nu_\pi^*}$ is controlled. By symmetry, we can perform the same procedures on P2 part and obtain an upper bound on $V^{\pi_\nu^*, \nu} - V^{\pi_{\nu^*}^*, \nu^*}$. Combining these together, we show sublinear regret rate of $V^{\pi_{\nu^*}^*, \nu^*} - V^{\pi^*, \nu_{\pi^*}^*}$. By regret to PAC conversion, this means that we find the policies that is the best achievable in $\Pi$.

---

**Algorithm 2** Alternate Optimistic Value Elimination (AOVE)

---

1: **Input:** Function class $\mathcal{F}$, policy class $\Pi$
2: Initialize $\mathcal{B}^0 \leftarrow \Pi \times \mathcal{F}$
3: Set $\beta \leftarrow C\log[\mathcal{N}(\mathcal{F}, 1/K) \cdot \mathcal{N}(\Pi, 1/K) \cdot HK/p]$ for some large constant $C$
4: **for** $k = 1, 2, \ldots, K$ **do**
5:      Find $(\pi^k, f^k) \leftarrow \text{argmax}_{(\pi,f)\in\mathcal{V}^{k-1}} f_1(x_1, \pi, \nu_\pi^f)$
6:      Let $g \leftarrow \underset{g:(\pi^k,g)\in\mathcal{V}^{k-1}}{\text{argmin}} g_1(x_1, \pi^k, \nu_{\pi^k}^g)$ and set $\nu^k \leftarrow \nu_{\pi^k}^g$
7:      **for** step $h = 1, 2, \ldots, H$ **do**
8:          P1 takes action $a_h^k \sim \pi_h^k(x_h^k)$, P2 takes action $b_h^k \sim \nu_h^k(x_h^k)$
9:          Observe reward $r_h^k$ and move to next state $x_{h+1}^k$
10:      **end for**
11:      Update $\mathcal{B}_h \leftarrow \mathcal{B}_h \cup \{(x_h^k, a_h^k, b_h^k, r_h^k, x_{h+1}^k)\}, \forall h \in [H]$
12:      For all $\xi, \zeta \in \mathcal{F}, \pi \in \Pi$ and $h \in [H]$, let

$$\mathcal{E}_{\mathcal{B}_h}(\xi_h, \zeta_{h+1}, \pi) = \sum_{\tau=1}^k [\xi_h(x_h^\tau, a_h^\tau, b_h^\tau) - r_h^\tau - \zeta_{h+1}(x_{h+1}^\tau, \pi, \nu_\pi^{\zeta_{h+1}})]^2$$

13:      Update

$$\mathcal{V}^k \leftarrow \{(\pi, f) \in \Pi \times \mathcal{F} : \mathcal{E}_{\mathcal{B}_h}(f_h, f_{h+1}, \pi) \leq \inf_{g\in\mathcal{F}_h} \mathcal{E}_{\mathcal{B}_h}(g, f_{h+1}, \pi) + \beta, \forall h \in [H]\}$$

14: **end for**

---

### 4.2.2 THEORETICAL RESULTS

This section presents our theoretical guarantees of Algorithm 2. First, we introduce two assumptions that take the policy class $\Pi$ into consideration. Variants of these assumptions are also common in batch RL (Jin et al., 2021c; Xie et al., 2021).

**Assumption 4.1** ($\Pi$-realizability). *For all $\pi \in \Pi$, $Q^{\pi, \nu_\pi^*} \in \mathcal{F}$.*

**Assumption 4.2** ($\Pi$-completeness). *For all $h \in [H]$, $\pi \in \Pi$, and $f \in \mathcal{F}_{h+1}$, $\mathcal{T}_h^\pi f \in \mathcal{F}_h$ holds.*

Notice that Assumption 4.2 reduces to $Q^*$ realizability in MDP. Both two assumptions hold for linear-parameterized Markov games studied in Xie et al. (2020). In fact, Xie et al. (2020) satisfies a stronger property: for any policy pair $\pi, \nu$ and any $h \in [H]$, there exists a vector $w \in \mathbb{R}^d$ such that

$$Q_h^{\pi, \nu}(x, a, b) = w^\top \phi(x, a, b), \ \forall(x, a, b) \in \mathcal{S} \times \mathcal{A}_1 \times \mathcal{A}_2.$$

Now we present the theory of Algorithm 2. First, we introduce the complexity measure in coordinated setting, which is a variant of Minimax Eluder dimension by replacing the Bellman operator Eq (1) with Eq (2). This Minimax Eluder dimension also allows a distribution family induced by function class $\mathcal{F}$ which is cheap to evaluate in large state space (Jin et al., 2021a).

**Definition 4.3** (Minimax Eluder dimension in Coordinated setting). *Let $(I - \mathcal{T}_h^{\Pi_{h+1}})\mathcal{F} := \{f_h - \mathcal{T}_h^\pi f_{h+1} : f \in \mathcal{F}, \pi \in \Pi_{h+1}\}$ be the class of residuals induced by the Bellman operator $\mathcal{T}_h^\pi$. Consider the following distribution families: 1) $\mathcal{D}_\Delta = \{\mathcal{D}_{\Delta,h}\}_{h\in[H]}$ where $\mathcal{D}_{\Delta,h} := \{\delta(x, a, b) : x \in \mathcal{S}, a \in \mathcal{A}_1, b \in \mathcal{A}_2\}$ is the set of Dirac measures on state-action*

*pair $(x, a, b)$; 2) $\mathcal{D}_\mathcal{F} = \{\mathcal{D}_{\mathcal{F},h}\}_{h \in [H]}$ which is generated by executing $(\pi_f, \nu^g_{\pi_f})$ for $f, g \in \mathcal{F}$. The $\epsilon$-Minimax Eluder dimension (in coordinated setting) of $\mathcal{F}$ is defined by $\dim_{ME'}(\mathcal{F}, \epsilon) := \max_{h \in [H]} \min_{\mathcal{D} \in \{\mathcal{D}_\Delta, \mathcal{D}_\mathcal{F}\}} \dim_{DE}((I - \mathcal{T}^{\Pi_{h+1}}_h)\mathcal{F}, \mathcal{D}, \epsilon).*

Now we present our main theory of Algorithm 2, where we make use of a variant of Minimax Eluder dimension that depends on policy class $\Pi$. We use $\mathcal{N}(\Pi, \epsilon)$ to denote the covering number of $\Pi$ under metric $d(\pi, \pi') := \max_{h \in [H]} \max_{f \in \mathcal{F}_h, x \in \mathcal{S}, b \in \mathcal{A}} |f(x, \pi, b) - f(x, \pi', b)|$.

**Theorem 4.4.** *Suppose Assumption 4.1 and Assumption 4.2 holds. With probability at least $1 - p$ we have the regret (Eq. (7)) of Algorithm 2 upper bounded by*

$$\text{Reg}(K) \le O\left(H\sqrt{K \cdot d_{ME'} \cdot \log[HK\zeta]}\right)$$

*where $d_{ME'} = \dim_{ME'}(\mathcal{F}, \sqrt{1/K})$ and $\zeta = \mathcal{N}(\mathcal{F}, 1/K)\mathcal{N}(\Pi, 1/K)/p$.*

As shown in this theorem, the regret is sub-linear in number of episodes $K$ and Eluder dimension $d_{ME'}$ which match the regret bound in Markov decision process, e.g. in Jin et al. (2021a). Through regret-to-PAC conversion, with high probability the algorithm can find $\epsilon$-optimal policy with $O\left(H^2 K d_{ME'} \log[\mathcal{N}(\mathcal{F}, 1/K)\mathcal{N}(\Pi, 1/K)HK/p]/\epsilon^2\right)$ sample complexity. When the reward and transition kernel have linear structures, we also recover the $\widetilde{O}(\sqrt{d^3 K})$ regret of Xie et al. (2020).

Although our upper bound depends on the logarithm of covering number of $\Pi$, there are some cases when it is small. For example, when the value-function class $\mathcal{F}$ is finite and $\Pi$ is the induced policy class of $\mathcal{F}$ as defined by $\Pi = \{\pi_f, \nu^g_{\pi_f} : f, g \in \mathcal{F}\}$, then $\log \mathcal{N}_\Pi(\epsilon) = \log|\mathcal{F}|$ and the regret is $\widetilde{O}(H\sqrt{K d_{ME'} \log^2 |\mathcal{F}|})$. Similarly, if the model class $\mathcal{M}$ is finite and $\Pi$ is induced policy class of $\mathcal{M}$ as defined by $\Pi = \{\pi_M, \nu^{M'}_{\pi_M} : M, M' \in \mathcal{M}\}$, then $\log \mathcal{N}_\Pi(\epsilon) = \log|\mathcal{M}|$ and the regret is $\widetilde{O}(H\sqrt{K d_{ME'} \log^2 |\mathcal{M}|})$. In agnostic setting which allows $\Pi$ to have arbitrary structure, the term $\log \mathcal{N}(\Pi, 1/K)$ reflects the difficulty of learning with complex candidate policy sets and we conjecture that this term is inevitable.

Similar to Algorithm 1, Algorithm 2 solves a non-convex optimization problem in Line 5-6 with highly non-convex constraint set (Line 13). This step is particularly difficult when the function class has complex structures. Algorithm 2 is in general computationally inefficient, even in cases of linear function approximations. Indeed, even when the model is known, solving for the NE is PPAD complete. Developing decentralized and provably efficient RL algorithm for multi-agent Markov game seems a challenging but interesting future direction.

## 5 DISCUSSION

In this paper we study function approximations in zero-sum simultaneous move Markov games. We design sample efficient algorithms for both decoupled and coordinated learning settings and propose Minimax Eluder dimension to characterize the complexity of decision making with value function approximation. The analysis shows $\sqrt{T}$ regret bounds and half-order dependence on the Minimax Eluder dimension, matching the results in MDPs and improving the linear function approximations. Our algorithms for coordinated setting are based on the idea of the 'alternate optimism', which also shows applicability in model-based methods.

### ACKNOWLEDGEMENTS

BH is supported by the Elite Undergraduate Training Program of School of Mathematical Sciences at Peking University. JDL acknowledges support of the ARO under MURI Award W911NF-11-1-0304, the Sloan Research Fellowship, NSF CCF 2002272, NSF IIS 2107304, and an ONR Young Investigator Award.

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

## A   ADDITIONAL RELATED WORKS

There is a rich literature on sample complexity for single agent RL, mostly focusing on the tabular cases (Strehl et al., 2006; Jaksch et al., 2010; Azar et al., 2017; Jin et al., 2018; Russo, 2019; Zanette & Brunskill, 2019) and the linear function approximations (Yang & Wang, 2020; Wang et al., 2021; Abbasi-Yadkori et al., 2019; Jin et al., 2020; Du et al., 2019b). For neural network function approximations, Agarwal et al. (2020); Liu et al. (2019); Wang et al. (2020a); Yang et al. (2020) builds provably efficient algorithms on over-parameterized models. The issue is complicated for general function approximation, as the Q function is hard to learn due to the double-sample issue (Baird, 1995). In recent years, a line of work studies the structural properties and corresponding complexity measures that allow generalization in RL. These work can be roughly grouped with two catagories. One studies structures with low information gain. Jiang et al. (2017) proposes Bellman rank and designs a sample efficient algorithm for a large family of MDPs. Later, Sun et al. (2019b) proposes Witness rank and uses this notion to provide a provably efficient model-based RL algorithm. Recently, Du et al. (2021) proposes Bilinear family and the corresponding Bilinear rank that encapsulates a wide variety of structural properties, spanning from both model-based to model-free function approximations. The other studies structures with low Eluder dimension (Russo & Van Roy, 2013). Osband & Van Roy (2014) extends Eluder dimension to reinforcement learning and provides a unified analysis of model based methods where the regret is controlled by the Kolmogorov dimension and Eluder dimension of model function class. Wang et al. (2020b) designs a model-free algorithm and shows that regret can be controlled by the Eluder dimension of the value function class. Recently, Jin et al. (2021a) proposes Bellman Eluder dimension and builds a new framework to capture a large amount of structures. It is noted that although lots of structures, such as linear MDP (Jin et al., 2020), fit in both two viewpoints, they are not equivalent. Among the above work, Jin et al. (2021a); Zanette et al. (2020) are more related. The method of placing uncertainty quantification on only initial states, previously appeared in Zanette et al. (2020); Jin et al. (2021a), greatly simplifies the argument and improves sample complexity. The Minimax Eluder dimension is inspired by the Bellman Eluder dimension in Jin et al. (2021a).

In competitive RL, the setting with access to a sampling oracle or well explored policies is well studied (Littman & Szepesvari, 1996; Greenwald et al., 2003; Grau-Moya et al., 2018; Pérolat et al., 2018; Srinivasan et al., 2018; Sidford et al., 2020). Under these assumptions, many works consider function approximations (Lagoudakis & Parr, 2002; Pérolat et al., 2015; Fan et al., 2020; Jia et al., 2019; Zhao et al., 2021). However, without strong sampling model or a well explored policy, the issue of exploration-exploitation tradeoff must be addressed. Most of these work focus on tabular setting (Wei et al., 2017; Pérolat et al., 2017; Bai & Jin, 2020; Bai et al., 2021; 2020; Zhang et al., 2020; Zhao et al., 2021) or linear function approximation settings (Xie et al., 2020; Chen et al., 2021). Among them, (Wei et al., 2017; Xie et al., 2020) are more related. The regret-decomposition method used in the decoupled setting is inspired by Xie et al. (2020). The 'alternate optimism' used in Algorithm 4 was previously used as 'Maximin-EVI' in Wei et al. (2017), although the 'alternate optimism' used in Algorithm 2 is different. Moreover, the concurrent work of Jin et al. (2021b) studies competitive RL with general function approximation. They reach a sublinear $\widetilde{O}(\sqrt{K})$ regret for MGs with low minimax Eluder dimensions in the decoupled setting. Their results in the coordinated setting are based on different assumptions and complexity measures. Specifically, they assume finite function class and an additional class of exploiters. It is noted that by letting the policy class be the class of optimal policies induced by the value function class, we have $|\Pi| = |\mathcal{F}|$ and our algorithm achieves on-par guarantee in their setup in terms of the covering numbers of function classes. In addition, we propose a first provably sample efficient and model-based algorithm for the coordinated setting with general function approximation.

### A.1   TECHNICAL CHALLENGES

Previous work (Xie et al., 2020) imposes optimistic bonus on the action-value functions in every state-action pairs and performs planning by the Coarse Correlated Equilibrium (CCE) on the optimistic value functions. To achieve improved rates, we leverage the idea of 'global optimism' (Zanette et al., 2020; Jin et al., 2021a; Du et al., 2021), which maintains a constraint set of candidate functions that do not deviate much from the empirical estimates and performs optimistic planning on the initial state. However, going beyond MDPs towards MGs, two problems arise. First, the concentration property of functions in constraint set is hard to characterize due to multi-agent interplay. For this,

we use the concentration methods in Jin et al. (2021a) and extend it from MDPs to MGs. The second and more prominent issue is the exploration and exploitation tradeoff. Since 'global optimism' only obtains optimism along the trajectories of behaviour policies, it may not guarantee optimism on the trajectories of target policies (i.e. NE). As a result, directly using CCE to plan will cause the duality gaps to diverge. To deal with this problem, we apply 'alternate optimism' to guide explorations, which was previously used in Wei et al. (2017) for model-based methods. The 'alternate optimism' used in this work is slightly different for value-based methods. We prove two regret decomposition lemmata to support this optimism principle.

## B  PROOF OF THEOREM 3.6

We prove the Theorem 3.6 in the following five steps. In the first step, we show that Bellman error is small for function in the constraint set. In the second step, we show that $Q^*$ stays in the constraint set throughout the algorithm. In the third step, we decompose the regret into a summation of Bellman error and Martingale difference. We can then bound the summation of Bellman error using Minimax Eluder dimension in step 4. Finally we combine the aforementioned steps and complete the proof of Theorem 3.6.

We define 'optimistic' value function in step $h$ and episode $k$ as follows

$$V_h^k(x_h^k) := f_h^k(x_h^k, \pi_h^k, \widehat{\nu}_h^k)$$

where $\widehat{\nu}_h^k$ is the best response of $\pi_h^k$ in value function $f^k$, i.e., $\widehat{\nu}_h^k := \operatorname{argmin}_\nu f_h^k(x_h^k, \pi_h^k, \nu)$. We use $\nu^k = \{\nu_h^k\}_{h=1}^H$ to denote the policy adopted by P2 in the $k$-th episode. Notice that the agent obtains knowledge from $\nu$ only through its actions $b_h^k, h \in [H]$.

**Step 1: Low Bellman error in the constraint set**    We first use a standard concentration procedure (Agarwal et al., 2012; Jin et al., 2021a) to bound the Bellman error in constraint sets.

**Lemma B.1** (Concentration on Bellman error). *Let $\rho > 0$ be an arbitrary fixed number. With probability at least $1 - p$ for all $(k, h) \in [K] \times [H]$ we have*

$$\sum_{i=1}^{k-1} \left( f_h^k(x_h^i, a_h^i, b_h^i) - r_h^i - \mathop{\mathbb{E}}_{x \sim \mathbb{P}(\cdot|x_h^i, a_h^i, b_h^i)} f_{h+1}^k(x, \pi_{h+1}^k, \widehat{\nu}_{h+1}^k) \right)^2 \le O(\beta).$$

*Proof.* Consider a fixed $(k, h, f)$ sample, let

$$U_t(h, f) := \left( f_h(x_h^t, a_h^t, b_h^t) - r_h^t - \min_{\nu'} \max_{\pi'} f_{h+1}(x_{h+1}^t, \pi', \nu') \right)^2$$

$$- \left( \mathcal{T}_h f_{h+1}(x_h^t, a_h^t, b_h^t) - r_h^t - \min_{\nu'} \max_{\pi'} f_{h+1}(x_{h+1}^t, \pi', \nu') \right)^2$$

and $\mathscr{F}_{t,h}$ be the filtration induced by $\{x_1^i, a_1^i, b_1^i, r_1^i, \ldots, x_H^i\}_{i=1}^{t-1} \cup \{x_1^t, a_1^t, b_1^t, r_1^t, \ldots, x_h^t, a_h^t, b_h^t\}$. Recall the minimax Bellman operator

$$\mathcal{T}_h f_{h+1}(x, a, b) := r(x, a, b) + \mathop{\mathbb{E}}_{x' \sim \mathbb{P}_h(\cdot|x,a,b)} [\min_{\nu'} \max_{\pi'} f_{h+1}(x', \pi', \nu')].$$

We have

$$\mathbb{E}[U_t(h, f)|\mathscr{F}_{t,h}]$$
$$= [(f_h - \mathcal{T}_h f_{h+1})(x_h^t, a_h^t, b_h^t)]$$
$$\quad \cdot \mathbb{E}[f_h(x_h^t, a_h^t, b_h^t) + \mathcal{T}_h f_{h+1}(x_h^t, a_h^t, b_h^t) - 2r_h^t - 2\min_{\nu'}\max_{\pi'} f_{h+1}(x_{h+1}^t, \pi', \nu')|\mathscr{F}_{t,h}]$$
$$= [(f_h - \mathcal{T}_h f_{h+1})(x_h^t, a_h^t, b_h^t)]^2$$

and

$$\operatorname{Var}[U_t(h, f)|\mathscr{F}_{t,h}] \le \mathbb{E}[(U_t(h, f))^2|\mathscr{F}_{t,h}]$$
$$= [(f_h - \mathcal{T}_h f_{h+1})(x_h^t, a_h^t, b_h^t)]^2$$
$$\quad \cdot \mathbb{E}[\left( f_h(x_h^t, a_h^t, b_h^t) + \mathcal{T}_h f_{h+1}(x_h^t, a_h^t, b_h^t) - 2r_h^t - 2\min_{\nu'}\max_{\pi'} f_{h+1}(x_{h+1}^t, \pi', \nu') \right)^2|\mathscr{F}_{t,h}]$$
$$\le 36[(f_h - \mathcal{T}_h f_{h+1})(x_h^t, a_h^t, b_h^t)]^2 = 36\,\mathbb{E}[U_t(h, f)|\mathscr{F}_{t,h}].$$

By Freedman's inequality, we have with probability at least $1 - p$,

$$\left| \sum_{t=1}^{k} U_t(h, f) - \sum_{t=1}^{k} \mathbb{E}[U_t(h, f)|\mathscr{F}_{t,h}] \right| \le O\left( \sqrt{\log(1/p) \sum_{t=1}^{k} \mathbb{E}[U_t|\mathscr{F}_{t,h}]} + \log(1/p) \right).$$

Let $\mathcal{N}(\mathcal{F}, \rho)$ be a $\rho$-cover of $\mathcal{F}$. Taking a union bound for all $(k, h, g) \in [K] \times [H] \times \mathcal{N}(\mathcal{F}, \rho)$, then with probability at least $1 - p$ for all $(k, h, g) \in [K] \times [H] \times \mathcal{N}(\mathcal{F}, \rho)$

$$\left| \sum_{t=1}^{k} U_t(h, g) - \sum_{t=1}^{k} [(g_h - \mathcal{T}_h g_{h+1})(x_h^t, a_h^t, b_h^t)]^2 \right|$$

$$\le O\left( \sqrt{\iota \sum_{t=1}^{k} [(g_h - \mathcal{T}_h g_{h+1})(x_h^t, a_h^t, b_h^t)]^2 + \iota} \right) \tag{8}$$

where $\iota = \log(HK|\mathcal{N}(\mathcal{F}, \rho)|/p)$. Let $g^k = \operatorname{argmin}_{g \in \mathcal{N}(\mathcal{F}, \rho)} \max_{h \in [H]} \|f_h^k - g_h^k\|_\infty$. We immediately have

$$\left| \sum_{t=1}^{k} U_t(h, g^k) - \sum_{t=1}^{k} [(g_h^k - \mathcal{T}_h g_{h+1}^k)(x_h^t, a_h^t, b_h^t)]^2 \right|$$

$$\le O\left( \sqrt{\iota \sum_{t=1}^{k} [(g_h^k - \mathcal{T}_h g_{h+1}^k)(x_h^t, a_h^t, b_h^t)]^2 + \iota} \right). \tag{9}$$

For all $(h, k) \in [H] \times [K]$, by the definition of $\mathcal{V}^k$ and $f^k \in \mathcal{V}^k$ we have

$$\sum_{t=1}^{k} U_t(h, f^k) = \sum_{t=1}^{k} \left( f_h^k(x_h^t, a_h^t, b_h^t) - r_h^t - \min_{\nu'} \max_{\pi'} f_{h+1}(x_{h+1}^t, \pi', \nu') \right)^2$$

$$- \sum_{t=1}^{k} \left( \mathcal{T}_h f_h^k(x_h^t, a_h^t, b_h^t) - r_h^t - \min_{\nu'} \max_{\pi'} f_{h+1}(x_{h+1}^t, \pi', \nu') \right)^2$$

$$\le \sum_{t=1}^{k} \left( f_h^k(x_h^t, a_h^t, b_h^t) - r_h^t - \min_{\nu'} \max_{\pi'} f_{h+1}(x_{h+1}^t, \pi', \nu') \right)^2$$

$$- \inf_{g \in \mathcal{F}} \sum_{t=1}^{k} \left( g_h(x_h^t, a_h^t, b_h^t) - r_h^t - \min_{\nu'} \max_{\pi'} f_{h+1}(x_{h+1}^t, \pi', \nu') \right)^2$$

$$\le O(\iota + k\rho). \tag{10}$$

It thus follows that,

$$\sum_{t=1}^{k} [(f_h^k - \mathcal{T}_h f_{h+1}^k)(x_h^t, a_h^t, b_h^t)]^2 \le \sum_{t=1}^{k} [(g_h^k - \mathcal{T}_h g_{h+1}^k)(x_h^t, a_h^t, b_h^t)]^2 + O(\iota + k\rho)$$

$$\le O(\sum_{t=1}^{k} U_t(h, g^k) + \iota + k\rho)$$

$$\le O(\sum_{t=1}^{k} U_t(h, f^k) + \iota + k\rho)$$

$$\le O(\iota + k\rho)$$

where the first step is due to $g^k$ is an $\rho$-approximation to $f^k$, the second step comes from Eq (9), the third step is due to $g^k$ is an $\rho$-approximation to $f^k$, and the final step comes from Eq (10).

By definition of $\pi^k$ and $\widehat{\nu}^k$, we have

$$\mathcal{T}_h f_{h+1}^k(x, a, b) := r(x, a, b) + \mathop{\mathbb{E}}_{x' \sim \mathbb{P}_h(\cdot|x,a,b)} [f_{h+1}(x', \pi_{h+1}^k, \widehat{\nu}_{h+1}^k)]$$

thus we conclude the proof. □

**Step 2: $Q^*$ is always in constraint set.**

**Lemma B.2.** *With probability at least $1 - p$, we have $Q^* \in \mathcal{V}^k$ for all $k \in [K]$.*

*Proof.* Consider a fixed $(k, h, f)$ sample, let

$$
U_t(h, f) := \left( f_h(x_h^t, a_h^t, b_h^t) - r_h^t - \min_{\nu'} \max_{\pi'} Q_{h+1}^*(x_{h+1}^t, \pi', \nu') \right)^2
$$
$$
- \left( Q_h^*(x_h^t, a_h^t, b_h^t) - r_h^t - \min_{\nu'} \max_{\pi'} Q_{h+1}^*(x_{h+1}^t, \pi', \nu') \right)^2
$$

and $\mathscr{F}_{t,h}$ be the filtration induced by $\{x_1^i, a_1^i, b_1^i, r_1^i, \ldots, x_H^i\}_{i=1}^{t-1} \cup \{x_1^t, a_1^t, b_1^t, r_1^t, \ldots, x_h^t, a_h^t, b_h^t\}$.

We have

$$
\mathbb{E}[U_t(h, f)|\mathscr{F}_{t,h}] = [(f_h - Q_h^*)(x_h^t, a_h^t, b_h^t)]^2
$$

and

$$
\mathrm{Var}[U_t(h, f)|\mathscr{F}_{t,h}] \le 36[(f_h^k - \mathcal{T}_h f_{h+1}^k)(x_h^t, a_h^t, b_h^t)]^2 = 36\,\mathbb{E}[U_t(h, f)|\mathscr{F}_{t,h}].
$$

Let $\mathscr{N}(\mathcal{F}, \rho)$ be a $\rho$-cover of $\mathcal{F}$. By Freedman's inequality, for all $(k, h, f) \in [K] \times [H] \times \mathscr{N}(\mathcal{F}, \rho)$ we have with probability at least $1 - p$,

$$
\left| \sum_{t=1}^k U_t(h, f) - \sum_{t=1}^k [(f_h - Q_h^*)(x_h^t, a_h^t, b_h^t)]^2 \right|
$$
$$
\le O\left( \sqrt{\log(1/p) \sum_{t=1}^k [(f_h - Q_h^*)(x_h^t, a_h^t, b_h^t)]^2} + O(\iota) \right)
$$

where $\iota = \log(HK|\mathscr{N}(\mathcal{F}, \rho)|/p)$. Since $[(f_h - Q_h^*)(x_h^t, a_h^t, b_h^t)]^2 \ge 0$, this implies

$$
-\sum_{t=1}^k U_t(h, g) \le O(\iota).
$$

Therefore for all $(k, h, g) \in [K] \times [H] \times \mathscr{N}(\mathcal{F}, \rho)$

$$
\sum_{t=1}^k \left( Q_h^*(x_h^t, a_h^t, b_h^t) - r_h^t - \min_{\nu'} \max_{\pi'} Q_{h+1}^*(x_{h+1}^t, \pi', \nu') \right)^2
$$
$$
\le \sum_{t=1}^k \left( f_h(x_h^t, a_h^t, b_h^t) - r_h^t - \min_{\nu'} \max_{\pi'} Q_{h+1}^*(x_{h+1}^t, \pi', \nu') \right)^2 + O(\iota + k\rho).
$$

Thus we have proven that $Q^* \in \mathcal{V}^k, \forall k \in [K]$ with probability at least $1 - p$.

$\square$

**Remark B.3.** *This step implies that*

$$
V^* \le V_1^k(x_1^k) \tag{11}
$$

*which ensures optimism from the beginning state ('global optimism').*

**Step 3: Regret decomposition** The following result is key to our analysis. It decomposes the deviation from the value of the game into Bellman errors across time steps and a martingale difference sequence.

**Lemma B.4.** *Define*

$$
\delta_h^k := V_h^k(x_h^k) - V_h^{\pi^k, \nu^k}(x_h^k)
$$
$$
\zeta_h^k := \mathbb{E}[\delta_{h+1}^k | x_h^k, a_h^k, b_h^k] - \delta_{h+1}^k
$$
$$
\gamma_h^k := \underset{a \sim \pi_h^k(x_h^k)}{\mathbb{E}}[f_h^k(x_h^k, a, b_h^k)] - f_h^k(x_h^k, a_h^k, b_h^k)
$$
$$
\widehat{\gamma}_h^k := \underset{a \sim \pi_h^k(x_h^k), b \sim \nu_h^k(x_h^k)}{\mathbb{E}}[Q_h^{\pi^k, \nu^k}(x_h^k, a, b)] - Q_h^{\pi^k, \nu^k}(x_h^k, a_h^k, b_h^k)
$$

*Then we have*

$$\delta_h^k \leq \delta_{h+1}^k - \zeta_h^k + \gamma_h^k - \widehat{\gamma}_h^k + \epsilon_h^k \tag{12}$$

*where $\epsilon_h^k$ is the bellman error defined by*

$$\epsilon_h^k := f_h^k(x_h^k, a_h^k, b_h^k) - r_h^k - \mathop{\mathbb{E}}_{x \sim \mathbb{P}(\cdot|x_h^k, a_h^k, b_h^k)} f_{h+1}^k(x, \pi_{h+1}^k, \widehat{\nu}_{h+1}^k).$$

*Proof.* By definition of $\widehat{\nu}_h^k = \arg\min_\nu f_h^k(x_h^k, \pi_h^k, \nu)$, we have

$$f_h^k(x_h^k, \pi_h^k, \widehat{\nu}_h^k) \leq f_h^k(x_h^k, \pi_h^k, b_h^k)$$
$$= f_h^k(x_h^k, a_h^k, b_h^k) + \gamma_h^k.$$

It thus follows that

$$\delta_h^k \leq f_h^k(x_h^k, a_h^k, b_h^k) - Q_h^{\pi^k, \nu^k}(x_h^k, a_h^k, b_h^k) + \gamma_h^k - \widehat{\gamma}_h^k$$
$$= r_h^k + \mathop{\mathbb{E}}_{x \sim \mathbb{P}(\cdot|x_h^k, a_h^k, b_h^k)} f_{h+1}^k(x, \pi_{h+1}^k, \widehat{\nu}_{h+1}^k) + \epsilon_h^k$$
$$- (r_h^k + \mathop{\mathbb{E}}_{x \sim \mathbb{P}(\cdot, x_h^k, a_h^k, b_h^k)}[V_{h+1}^{\pi^k, \nu^k}(x)|x_h^k, a_h^k, b_h^k]) + \gamma_h^k - \widehat{\gamma}_h^k$$
$$= f_h^k(x_{h+1}^k, \pi_{h+1}^k, \widehat{\nu}_{h+1}^k) - V_{h+1}^{\pi^k, \nu^k}(x_{h+1}^k) - \zeta_h^k + \epsilon_h^k + \gamma_h^k - \widehat{\gamma}_h^k$$
$$= \delta_{h+1}^k - \zeta_h^k + \gamma_h^k - \widehat{\gamma}_h^k + \epsilon_h^k.$$

Therefore we complete the proof. $\qquad\square$

**Step 4: Bounding cumulative Bellman error using Minimax Eluder dimension.** Recall that Lemma B.1 gives:

$$\sum_{i=1}^{k-1} \left( f_h^k(x_h^i, a_h^i, b_h^i) - r_h^i - \mathop{\mathbb{E}}_{x \sim \mathbb{P}(\cdot|x_h^i, a_h^i, b_h^i)} f_{h+1}^k(x, \pi_{h+1}^k, \widehat{\nu}_{h+1}^k) \right)^2 \leq O(\beta).$$

Combining this and Lemma F.1 with

$$g_k = f_h^k(\cdot, \cdot, \cdot) - r_h^k - \mathop{\mathbb{E}}_{x \sim \mathbb{P}(\cdot|\cdot, \cdot, \cdot)} f_{h+1}^k(x, \pi_{h+1}^k, \widehat{\nu}_{h+1}^k)$$

and $\rho_k = \delta(x_n^k, a_h^k, b_h^k)$, we have come to the following result.

**Lemma B.5.** *For any $(k, h) \in [K] \times [H]$ we have*

$$\sum_{k=1}^K |f_h^k(x_h^k, a_h^k, b_h^k) - r_h^k - \mathop{\mathbb{E}}_{x \sim \mathbb{P}(\cdot|x_h^k, a_h^k, b_h^k)} f_{h+1}^k(x, \pi_{h+1}^k, \widehat{\nu}_{h+1}^k)|$$
$$\leq O\left( \sqrt{K \cdot \dim_{\mathrm{ME}}(\mathcal{F}, \sqrt{1/K}) \log(HK\mathcal{N}(\mathcal{F}, 1/K)/p)} \right).$$

*Thus*

$$\sum_{k=1}^K \epsilon_h^k \leq O\left( \sqrt{K \cdot \dim_{\mathrm{ME}}(\mathcal{F}, \sqrt{1/K}) \log(HK\mathcal{N}(\mathcal{F}, 1/K)/p)} \right). \tag{13}$$

**Step 5: Putting everything together** By Eq (11) and Eq (12)

$$\sum_{k=1}^K [V_1^*(x_1) - V_1^{\pi^k, \nu^k}(x_1)] \leq \sum_{k=1}^K [V_1^k(x_1) - V_1^{\pi^k, \nu^k}(x_1)]$$
$$\leq \sum_{k=1}^K \sum_{h=1}^H (-\zeta_h^k + \gamma_h^k - \widehat{\gamma}_h^k + \epsilon_h^k)$$
$$= \sum_{h=1}^H \sum_{k=1}^K (-\zeta_h^k + \gamma_h^k - \widehat{\gamma}_h^k) + \sum_{h=1}^H \sum_{k=1}^K \epsilon_h^k.$$

Notice that $\sum_{h=1}^{H} \sum_{k=1}^{K} (-\zeta_h^k + \gamma_h^k - \widehat{\gamma}_h^k)$ is a martingale difference sequence and can be bounded by $O(H\sqrt{KH \log(HK/p)})$ with probability at least $1 - p$.

By Eq (13) the term $\sum_{h=1}^{H} \sum_{k=1}^{K} \epsilon_h^k$ can be bounded by

$$O\left( H\sqrt{K \cdot \dim_{\mathrm{ME}}(\mathcal{F}, \sqrt{1/K}) \log(HK\mathcal{N}(\mathcal{F}, 1/K)/p)} \right)$$

with probability at least $1 - p$.

It thus follows that regret can be bounded by

$$\mathrm{Reg}(K) \leq O\left( H\sqrt{K \cdot \dim_{\mathrm{ME}}(\mathcal{F}, \sqrt{1/K}) \log(HK\mathcal{N}(\mathcal{F}, 1/K)/p)} \right).$$

## B.1 LINEAR FUNCTION APPROXIMATION

In the linear function approximation setting of Xie et al. (2020),

$$r_h(x, a, b) = \phi(x, a, b)^\top \theta_h, \quad \mathbb{P}_h(\cdot|x, a, b) = \phi(x, a, b)^\top \mu_h(\cdot)$$

where $\phi(\cdot, \cdot, \cdot) \in \mathbb{R}^d$ is known feature vector and $\theta_h \in \mathbb{R}^d$ and $\mu_h(\cdot) \in \mathbb{R}^d$ are unknown with $\|\theta\|_2 \leq \sqrt{d}$, $\|\nu_h(\cdot)\|_2 \leq \sqrt{d}$ and $\|\phi(\cdot, \cdot)\|_2 \leq 1$. In this case Algorithm 1 reduces to the following Algorithm 3. Notice that this algorithm can also be seen as a generalization of Zanette et al. (2020) to Markov games.

---

**Algorithm 3** ONEMG for linear function approximation.

---

1: **Input:** Function class $\mathcal{F}$
2: Set $\beta \leftarrow C\sqrt{d} \log(HK/p)$ for some large constant $C$
3: **for** episode $k = 1, 2, \ldots, K$ **do**
4:     Receive initial state $x_1^k$
5:     Set $w_{H+1}^k = \theta_{H+1}^k = 0$
6:     Find $\max_{w_h^k, \theta_h^k, h \in [H]} V_1^k(x_1^k)$ such that for all $h \in [H]$:

$$
\begin{aligned}
&(1) \;\; w_h^k = (\Lambda_h^k)^{-1} \sum_{\tau=1}^{k-1} \phi(x_h^\tau, a_h^\tau, b_h^\tau)[r_h^\tau + V_{h+1}^k(x_{h+1}^\tau)] \\
&(2) \;\; \|\theta_h^k - w_h^k\|_{\Lambda_h^k} \leq C \cdot H\beta \\
&(3) \;\; Q_h^k(\cdot, \cdot, \cdot) = (\theta_h^k)^\top \phi(\cdot, \cdot, \cdot) \\
&(4) \;\; (\pi_h^k, B_0) = \mathrm{NE}(Q_h^k) \\
&(5) \;\; V_h^k(\cdot) = \mathbb{E}_{a \sim \pi_h^k, b \sim B_0} [Q_h^k(\cdot, a, b)]
\end{aligned}
$$

7:     **for** step $h = 1, 2, \ldots, H$ **do**
8:         P1 plays action $a_h^k \sim \pi_h^k(x_h^k)$
9:         P2 takes action $b_h^k$
10:        Observe reward $r_h^k$ and move to next state $x_{h+1}^k$
11:     **end for**
12: **end for**

---

We will prove the following theoretical result of Algorithm 3. Notice the dependency on $d$ of Algorithm 3 is $d$ while the dependency on $d$ of Xie et al. (2020) is $d^{3/2}$, thus this result improves theirs by $\sqrt{d}$ factor.

**Theorem B.6.** *Regret (Eq (3)) in Algorithm 3 is bounded by $O(d\sqrt{H^2 K \log(dHK/p)})$ with probability at least $1 - p$.*

### B.1.1 PROOF OF THEOREM B.6

We prove Theorem B.6 in the following five steps similar to the previous section.

**Step 1: Low Bellman error in the constraint set** We use some standard results from previous work. Notice that the $\mathcal{N}(\mathcal{F}, \epsilon) = O(d \log(1/\epsilon))$ and our inherent Bellman error is zero, thus these lemmata can be directly adapted into our setting.

**Lemma B.7** (Concentration, adapted from Lemma 1 of Zanette et al. (2020)). *With probability at least $1 - p$, the following holds for all $(k, h) \in [K] \times [H]$,*

$$\left\| \sum_{\tau \in [k-1]} \phi_h^\tau [V_{h+1}^k(x_{h+1}^\tau) - \mathbb{E}_{x \sim \mathbb{P}(\cdot | x_h^\tau, a_h^\tau, b_h^\tau)} V_{h+1}^k(x)] \right\|_{(\Lambda_h^k)^{-1}} \leq H\beta$$

*where $\beta = \Omega(\sqrt{d \log(dKH/p)})$.*

**Lemma B.8** (Least square error bound, adapted from Lemma 3 of Xie et al. (2020)). *On the event of Lemma B.7, the following holds for all $(x, a, b, h, k) \in \mathcal{S} \times \mathcal{A} \times \mathcal{A} \times [H] \times [K]$ and any policy pair $(\pi, \nu)$*

$$\left| \langle \phi(x, a, b), \theta_h^k \rangle - Q_h^{\pi, \nu}(x, a, b) - \mathbb{E}_{x \sim \mathbb{P}(\cdot | x, a, b)} (V_{h+1}^k - V_{h+1}^{\pi, \nu})(x) \right| \leq \beta \|\phi(x, a, b)\|_{(\Lambda_h^k)^{-1}}$$

**Step 2: $Q^*$ always stays in the constraint set**

**Lemma B.9** (Optimism). *Let $Q_h^*(\cdot, \cdot, \cdot) = \phi(\cdot, \cdot, \cdot)^\top \theta_h^*$, then on the event of Lemma B.8 and Lemma B.7 there exists $w_h^*, h \in [H]$ such that $w_h^*, \theta_h^*, h \in [H]$ satisfies the constraints of Line 6 in Algorithm 3.*

*Proof.* We prove by backward induction on $h$. The claim trivially holds for $h = H + 1$. Now assume the inductive hypothesis holds at $h + 1$. Let

$$w_h^* = (\Lambda_h^k)^{-1} \sum_{\tau=1}^{k-1} \phi(x_h^\tau, a_h^\tau, b_h^\tau)[r_h^\tau + V_{h+1}^k(x_{h+1}^\tau)]$$

$$= \left( \sum_{\tau=1}^{k-1} \phi_h^\tau (\phi_h^\tau)^\top + I \right)^{-1} \sum_{\tau=1}^{k-1} \phi_h^\tau \left( r_h^\tau + \mathbb{E}_{x \sim \mathbb{P}(\cdot | x_h^\tau, a_h^\tau, b_h^\tau)} V_{h+1}^k(x) + \eta_h^\tau \right)$$

where $\phi_h^\tau = \phi(x_h^\tau, a_h^\tau, b_h^\tau)$ and $\eta_h^\tau = V_{h+1}^k(x_{h+1}^\tau) - \mathbb{E}_{x \sim \mathbb{P}(\cdot | x_h^\tau, a_h^\tau, b_h^\tau)} V_{h+1}^k(x)$. Notice that inductive hypothesis implies $r_h^\tau + \mathbb{E}_{x \sim \mathbb{P}(\cdot | x_h^\tau, a_h^\tau, b_h^\tau)} V_{h+1}^k(x) = \phi(x_h^\tau, a_h^\tau, b_h^\tau)^\top \theta_h^*$, therefore

$$w_h^* = \left( \sum_{\tau=1}^{k-1} \phi_h^\tau (\phi_h^\tau)^\top + I \right)^{-1} \sum_{\tau=1}^{k-1} \phi_h^\tau \left( (\phi_h^\tau)^\top \theta_h^* + \eta_h^\tau \right)$$

$$= \theta_h^* - (\Lambda_h^k)^{-1} \theta_h^* + (\Lambda_h^k)^{-1} \sum_{\tau=1}^{k-1} \phi_h^\tau \eta_h^\tau.$$

It thus suffices to bound the following

$$\|(\Lambda_h^k)^{-1} \theta_h^* + (\Lambda_h^k)^{-1} \sum_{\tau=1}^{k-1} \phi_h^\tau \eta_h^\tau\|_{(\Lambda_h^k)} \leq \|\theta_h^*\|_{(\Lambda_h^k)^{-1}} + \|\sum_{\tau=1}^{k-1} \phi_h^\tau \eta_h^\tau\|_{(\Lambda_h^k)^{-1}}$$

$$\leq \sqrt{d} + H\beta$$

$$\leq C \cdot H\beta$$

where the penultimate step comes from $\|\theta_h^*\|_2 \leq \sqrt{d}$ and Lemma B.7. $\square$

**Remark B.10.** *This fact implies that*

$$V^*(x_1^k) \leq V_1^k(x_1^k). \tag{14}$$

**Step 3: Recursive decomposition of regret**

**Lemma B.11** (Regret decomposition). *Define*

$$\delta_h^k := V_h^k(x_h^k) - V_h^{\pi^k, \nu^k}(x_h^k)$$

$$\zeta_h^k := \mathbb{E}[\delta_{h+1}^k | x_h^k, a_h^k, b_h^k] - \delta_{h+1}^k$$

$$\gamma_h^k := \mathop{\mathbb{E}}_{a \sim \pi_h^k(x_h^k)}[Q_h^k(x_h^k, a, b_h^k)] - Q_h^k(x_h^k, a_h^k, b_h^k)$$

$$\widehat{\gamma}_h^k := \mathop{\mathbb{E}}_{a \sim \pi_h^k(x_h^k), b \sim \nu_h^k(x_h^k)}[Q_h^{\pi^k, \nu^k}(x_h^k, a, b)] - Q_h^{\pi^k, \nu^k}(x_h^k, a_h^k, b_h^k)$$

*Then we have*

$$\delta_h^k \le \delta_{h+1}^k - \zeta_h^k + \gamma_h^k - \widehat{\gamma}_h^k + \epsilon_h^k \tag{15}$$

*where $\epsilon_h^k$ is the width defined by*

$$\epsilon_h^k := 2\beta \sqrt{(\phi_h^k)^\top (\Lambda_h^k)^{-1} \phi_h^k}.$$

*Proof.* By definition of $B_0 = \mathrm{argmin}_\nu f_h^k(x_h^k, \pi_h^k, \nu)$, we have

$$V_h^k(x_h^k) = Q_h^k(x_h^k, \pi_h^k, B_0)$$
$$\le Q_h^k(x_h^k, \pi_h^k, b_h^k)$$
$$= Q_h^k(x_h^k, a_h^k, b_h^k) + \gamma_h^k.$$

It thus follows that

$$\delta_h^k \le Q_h^k(x_h^k, a_h^k, b_h^k) - Q_h^{\pi^k, \nu^k}(x_h^k, a_h^k, b_h^k) + \gamma_h^k - \widehat{\gamma}_h^k$$
$$\le \mathop{\mathbb{E}}_{x \sim \mathbb{P}(\cdot | x_h^k, a_h^k, b_h^k)}[V_{h+1}^k(x)] - \mathop{\mathbb{E}}_{x \sim \mathbb{P}(\cdot | x_h^k, a_h^k, b_h^k)}[V_{h+1}^{\pi^k, \nu^k}(x)] + \epsilon_h^k + \gamma_h^k - \widehat{\gamma}_h^k$$
$$= V_{h+1}^k(x_{h+1}^k) - V_{h+1}^{\pi^k, \nu^k}(x_{h+1}^k) - \zeta_h^k + \epsilon_h^k + \gamma_h^k - \widehat{\gamma}_h^k$$
$$= \delta_{h+1}^k - \zeta_h^k + \gamma_h^k - \widehat{\gamma}_h^k + \epsilon_h^k$$

where the second step comes from Lemma B.8 by letting $(x, a, b) = (x_h^k, a_h^k, b_h^k)$ (notice that $\langle \phi(x_h^k, a_h^k, b_h^k), \theta_h^k \rangle = Q_h^k(x_h^k, a_h^k, b_h^k)$). $\square$

**Step 4: Bounding width by Elliptical Potential Lemma** This step directly makes use of the following standard result.

**Lemma B.12** (Elliptical Potential Lemma, Lemma 10 of Xie et al. (2020)). *Suppose $\{\phi_t\}_{t \ge 0}$ is a sequence in $\mathbb{R}^d$ satisfying $\|\phi_t\| \le 1, \forall t$. Let $\Lambda_0 \in \mathbb{R}^{d \times d}$ be a positive definite matrix, and $\Lambda_t = \Lambda_0 + \sum_{i \in [t]} \phi_i \phi_i^\top$. If the smallest eigenvalue of $\Lambda_0$ is lower bounded by 1, then*

$$\log(\frac{\det \Lambda_t}{\det \Lambda_0}) \le \sum_{i \in [t]} \phi_i^\top \Lambda_{j-1}^{-1} \phi_i \le 2 \log(\frac{\det \Lambda_t}{\det \Lambda_0}).$$

**Step 5: Putting everything together** By Eq (14) and Eq (15)

$$\sum_{k=1}^K [V_1^*(x_1) - V_1^{\pi^k, \nu^k}(x_1)] \le \sum_{k=1}^K [V_1^k(x_1) - V_1^{\pi^k, \nu^k}(x_1)]$$
$$\le \sum_{k=1}^K \sum_{h=1}^H (-\zeta_h^k + \gamma_h^k - \widehat{\gamma}_h^k + \epsilon_h^k)$$
$$= \sum_{h=1}^H \sum_{k=1}^K (-\zeta_h^k + \gamma_h^k - \widehat{\gamma}_h^k) + \sum_{h=1}^H \sum_{k=1}^K \epsilon_h^k.$$

Notice that $\sum_{h=1}^{H} \sum_{k=1}^{K} (-\zeta_h^k + \gamma_h^k - \widehat{\gamma}_h^k)$ is a martingale difference sequence and can be bounded by $H\sqrt{KH\log(HK/p)}$ with probability at least $1 - p$.

By Lemma B.12 the term $\sum_{h=1}^{H} \sum_{k=1}^{K} \epsilon_h^k$ can be bounded as follows

$$
\begin{aligned}
\sum_{h=1}^{H} \sum_{k=1}^{K} \epsilon_h^k &= \sum_{h=1}^{H} \sum_{k=1}^{K} 2\beta \sqrt{(\phi_h^k)^\top (\Lambda_h^k)^{-1} \phi_h^k} \\
&\leq \sum_{h=1}^{H} \sqrt{K} \cdot \sqrt{\sum_{k=1}^{K} 2\beta (\phi_h^k)^\top (\Lambda_h^k)^{-1} \phi_h^k} \\
&\leq 2\beta \sum_{h=1}^{H} \sqrt{K} \cdot \sqrt{2\log(\frac{\det\Lambda_h^k}{\det\Lambda_h^0})} \\
&\leq 2\beta \sum_{h=1}^{H} \sqrt{K} \cdot \sqrt{2\log \frac{(\lambda + K\max_k \|\phi_h^k\|^2)^d}{\lambda^d}} \\
&\leq O(dH\sqrt{K\log(dHK/p)}).
\end{aligned}
$$

with probability at least $1 - p$.

It thus follows that with probability at least $1 - p$ regret can be upper bounded by

$$
O(d\sqrt{H^2 K\log(dHK/p)}).
$$

## C    MODEL-BASED METHOD FOR THE COORDINATED SETTING

### C.1    ALTERNATE OPTIMISTIC MODEL ELIMINATION (AOME)

---

**Algorithm 4** Alternate Optimistic Model Elimination (AOME)

---

1: **Input:** Model class $\mathcal{M}$, test function class $\mathcal{G}$, precision $\epsilon$, failure probability $p$
2: Initialize $\mathcal{M}^0 \leftarrow \mathcal{M}$
3: **for** $k = 1, 2, \ldots, K$ **do**
4:     Find $M_1^k \leftarrow \mathrm{argmax}_{M \in \mathcal{M}} Q^M(x_1, \pi^M, \nu^M)$, let $\pi^k \leftarrow \pi^{M_1^k}$
5:     Find $M_2^k \leftarrow \mathrm{argmin}_{M \in \mathcal{M}} Q^M(x_1, \pi^k, \nu_{\pi^k}^M)$, let $\nu^k \leftarrow \nu_{\pi^k}^{M_1^k}$
6:     Execute $\pi^k, \nu^k$ and collect $n_1$ rewards $\{r_h^i\}_{h \in [H], i \in [n_1]}$
7:     Let $\widehat{V}^k \leftarrow \frac{1}{n_1} \sum_{i=1}^{n_1} (\sum_{h=1}^{H} r_h^i)$
8:     **if** $\max\{|\widehat{V}^k - Q^{M_1^k}(x_1, \pi^k, \nu^k)|, |\widehat{V}^k - Q^{M_2^k}(x_1, \pi^k, \nu^k)|\} \leq \epsilon/2$ **then**
9:         Terminate and output $\pi^k, \nu^k$.
10:    **else**
11:        Find $h^k$ such that $\max_{i \in [2]} |\widehat{\mathcal{L}}(M_1^k, M_2^k, M_i^k, h^k)| \geq \epsilon/(4H)$ by Eq (5).
12:        Collect $n$ trajectories $\{(x_h^i, a_h^i, b_h^i, r_h^i)_{h=1}^H\}_{i=1}^n$ by executing $a_h, b_h \sim \pi_h^k, \nu_h^k, \forall h \in [H]$
13:        Update constraint set, where $\widehat{\mathcal{E}}$ is given by Eq (6)

$$
\mathcal{M}^k \leftarrow \{M \in \mathcal{M}^{k-1} : \widehat{\mathcal{E}}(M_1^k, M_2^k, M, h^k) \leq \phi\}
$$

14:    **end if**
15: **end for**

---

### C.2    THEORY FOR ALTERNATE OPTIMISTIC MODEL ELIMINATION (AOME)

In this section we prove that Algorithm 4 terminates in finite rounds. The technique bulk is then showing that in Step 13 the constraint set can only shrink for finite times. This is dependent on the complexity of model class $\mathcal{M}$ and discriminator function class $\mathcal{G}$. We first generalize Witness rank (Sun et al., 2019b) to Markov games.

**Definition C.1** (Witnessed model misfit in Markov games). *For discriminator class $\mathcal{G} : \mathcal{S} \times \mathcal{A}_1 \times \mathcal{A}_2 \times \mathbb{R} \times \mathcal{S} \mapsto \mathbb{R}$, models $M_1, M_2, M \in \mathcal{M}$ and level $h \in [H]$, the witnessed model misfit of Markov game at level $h$ is defined as follow*

$$\mathcal{E}(M_1, M_2, M, h) := \sup_{g \in \mathcal{G}} \mathbb{E}_{(x,a,b) \sim \mathcal{P}_{M_1}^{M_2}} \mathbb{E}_{(r,x) \sim M} [g(x_h, a_h, b_h, r, x) - g(x_h, a_h, b_h, r_h, x_{h+1})]. \quad (16)$$

*where $(x, a, b) \sim \mathcal{P}_{M_1}^{M_2}$ denotes $x_{\tau+1} \sim \mathbb{P}(\cdot|x_\tau, a_\tau, b_\tau)$, $a_\tau \sim \pi^{M_1}$ and $b_\tau \sim \nu_{\pi^{M_1}}^{M_2}$ for all $\tau \in [h]$.*

Using $g(x_h, a_h, b_h, r, x) = r(x_h, a_h, b_h) + Q_M(x, \pi^{M_1}, \nu_{\pi^{M_1}}^{M_2})$, we can also give a generalization of Bellman error (Jiang et al., 2016) in Markov games by the following

$$\mathcal{L}(M_1, M_2, M, h) := \mathbb{E}_{(x,a,b) \sim \mathcal{P}_{M_1}^{M_2}} \left[ \mathbb{E}_{(r,x) \sim M} [g(x_h, a_h, b_h, r, x))] - \mathbb{E}_{(r,x) \sim M^*} [g(x_h, a_h, b_h, r, x)] \right].$$
$$(17)$$

The final components in our theory are two assumptions from Sun et al. (2019b). Define $\text{rank}(A, \beta)$ to be the smallest integer $k$ such that $A = UV^\top$ with $U, V \in \mathbb{R}^{n \times k}$ and $\|U_{i,*}\|_2 \cdot \|V_{j,*}\|_2 \leq \beta$ for any pair of rows $U_{i,*}, V_{j,*}$.

**Assumption C.2** (Realizability). *Assume $M^* \in \mathcal{M}$.*

**Assumption C.3** (Witness misfit domination). *Assume $\mathcal{G}$ is finite, $\|g\|_\infty \leq 2, \forall g \in \mathcal{G}$ and $\forall M_1, M_2, M \in \mathcal{M} : \mathcal{E}(M_1, M_2, M, h) \geq \mathcal{L}(M_1, M_2, M, h)$.*

Now we introduce the Witness rank for Markov games.

**Definition C.4** (Witness rank for Markov games). *Given a model class $\mathcal{M}$, a test function class $\mathcal{G}$ and $\kappa \in (0, 1]$. For $h \in [H]$, we define $\mathcal{N}_{\kappa,h} \subset \mathbb{R}^{|\mathcal{M}|^2 \times |\mathcal{M}|}$ as the set of matrices such that any matrix $A \in \mathcal{N}_{\kappa,h}$ satisfies*

$$\kappa |\mathcal{L}(M_1, M_2, M, h)| \leq A((M_1, M_2), M) \leq \mathcal{E}(M_1, M_2, M, h), \ \forall M_1, M_2, M \in \mathcal{M}.$$

*The Witness rank for Markov games is defined by*

$$W(\kappa, \beta, \mathcal{M}, \mathcal{G}, h) := \min_{A \in \mathcal{N}_{\kappa,h}} \text{rank}(A, \beta).$$

We are in the position to state the main theorem of this section.

**Theorem C.5.** *Under Assumption C.2 and Assumption C.3, set $\phi = \frac{\kappa\epsilon}{100H\sqrt{W_\kappa}}$ where $W_\kappa = \max_{h \in [H]} W(\kappa, \beta, \mathcal{M}, \mathcal{G}, h)$, let $T = HW_\kappa \log(\beta/2\phi)$, set $n_1 = C \cdot H^2 \log(HT/p)/\epsilon^2$ and $n = C \cdot H^2 W_\kappa |\mathcal{A}| \log(T|\mathcal{M}||\mathcal{G}|/p)/(\kappa\epsilon)^2$ for large enough constant $C$. Then for all $\epsilon, p, \kappa \in (0, 1]$, with probability at least $1 - p$ Algorithm 4 outputs a policy $\pi$ such that $V^*(x_1) - V^{\pi,\nu_\pi^*}(x_1) \leq \epsilon$ within at most $O(\frac{H^3 W_\kappa^2 |\mathcal{A}|}{\kappa^2 \epsilon^2} \log(T|\mathcal{G}||\mathcal{M}|/p))$ trajectories.*

As seen in the above theorem, Algorithm 4 has sample complexity that scales quadratic with Witness rank and logarithmically with cardinality of model class. This matches the sample complexity results in the Markov decision process setting (Sun et al., 2019b). When the test function is Q-function, the Witnessed model misfit reduced to Bellman error as seen in Eq (17) and witness rank reduce to Bellman rank.

## C.3 PROOFS

We prove Theorem C.5 in the following steps. In the first step, we show a simulation lemma that generalize Jiang et al. (2017) to Markov games. In the second step we show concentration of empirical estimates of Bellman error, model misfit and value functions. In the third step we show that $M^*$ always stays in the constraint set. We examine the conditions when Algorithm 4 terminates or not in the fourth step. Using these conditions, we bound the number of rounds in the fifth step. Finally we combine everything and complete the proof of Theorem C.5.

From Definition C.4 we know that there exists $A_h \in \mathcal{N}_{\kappa,h}$ such that

$$\kappa |\mathcal{L}(M_1, M_2, M, h)| \leq A_h((M_1, M_2), M) \leq \mathcal{E}(M_1, M_2, M, h), \ \forall M_1, M_2, M \in \mathcal{M}.$$

and we can factorize $A_h((M_1, M_2), M) = \langle \zeta_h(M_1, M_2), \chi_h(M) \rangle$ where $\|\zeta_h(M_1, M_2)\|_2, \|\chi_h(M)\|_2 \leq \beta$.

**Step 1: Simulation Lemma**   The analysis begins with a useful Lemma that decouples the value difference into a sum of Bellman errors across time steps.

**Lemma C.6** (Simulation Lemma). *Fix model $M_1, M_2, M \in \mathcal{M}$. Let $\pi = \pi^{M_1}, \nu = \nu^{M_2}_{\pi^{M_1}}$. Under Assumption C.3, we have*

$$Q_M(x_1, \pi, \nu) - V_1^{\pi, \nu}(x_1) = \sum_{h=1}^{H} \mathcal{L}(M_1, M_2, M, h)$$

*Proof.* Let $(x, a, b) \sim \mathcal{P}^{M_2}_{M_1}$ denote $x_{\tau+1} \sim \mathbb{P}(\cdot | x_\tau, a_\tau, b_\tau)$, $a_\tau \sim \pi^{M_1}$ and $b_\tau \sim \nu^{M_2}_{\pi^{M_1}}$ for all $\tau \in [h]$. Using definition of Bellman error in Eq (17), we have

$$Q_M(x_1, \pi, \nu) - V_1^{\pi, \nu}(x_1)$$

$$= \mathop{\mathbb{E}}_{(x,a,b)\sim\mathcal{P}^{M_2}_{M_1}} \left[ \mathop{\mathbb{E}}_{(r_1, x_2)\sim M}[r_1 + Q_M(x_2, \pi, \nu)] - \mathop{\mathbb{E}}_{(r_1, x_2)\sim M^*}[r_1 + Q_{M^*}(x_2, \pi, \nu)] \right]$$

$$= \mathop{\mathbb{E}}_{(x,a,b)\sim\mathcal{P}^{M_2}_{M_1}} \left[ \mathop{\mathbb{E}}_{(r_1, x_2)\sim M}[r_1 + Q_M(x_2, \pi, \nu)] - \mathop{\mathbb{E}}_{(r_1, x_2)\sim M^*}[r_1 + Q_M(x_2, \pi, \nu)] \right]$$

$$+ \mathop{\mathbb{E}}_{(x,a,b)\sim\mathcal{P}^{M_2}_{M_1}} \left[ \mathop{\mathbb{E}}_{(r_1, x_2)\sim M^*}[r_1 + Q_M(x_2, \pi, \nu)] - \mathop{\mathbb{E}}_{(r_1, x_2)\sim M^*}[r_1 + Q_{M^*}(x_2, \pi, \nu)] \right]$$

$$= \mathcal{L}(M_1, M_2, M, 1) + \mathop{\mathbb{E}}_{(x,a,b)\sim\mathcal{P}^{M_2}_{M_1}}[Q_M(x_2, \pi, \nu) - V_2^{\pi, \nu}(x_2)]$$

$$= \cdots$$

$$= \sum_{h=1}^{H} \mathcal{L}(M_1, M_2, M, h).$$

We thus complete the proof. $\qquad\square$

**Step 2: Concentrations**   In this part we show some standard concentration results.

**Lemma C.7.** *With probability at least $1 - p$, we have the following event*

1. $\left| \mathcal{E}(M_1^k, M_2^k, M, h) - \widehat{\mathcal{E}}(M_1^k, M_2^k, M, h) \right| \leq \phi$ *for all $M \in \mathcal{M}$ and $h \in [H]$,*

2. $\left| \widehat{\mathcal{L}}(M_1^k, M_2^k, M_i^k, h) - \mathcal{L}(M_1^k, M_2^k, M_i^k, h) \right| \leq \frac{\epsilon}{8H}$ *for all $h \in [H]$ and $i \in [2]$,*

3. $\left| V^{\pi^k, \nu^k} - \widehat{V}^k \right| \leq \epsilon/8$

*holds for all $k \leq T$ rounds.*

*Proof.* Fix one iteration. By Hoeffding's inequality, with probability at least $1 - p/3$,

$$\left| \widehat{\mathcal{L}}(M_1^k, M_2^k, M_i^k, h) - \mathcal{L}(M_1^k, M_2^k, M_i^k, h) \right| \leq \sqrt{\frac{\log(2H/p)}{n_1}}. \qquad (18)$$

Thus (2) follows from $n_1 = C \cdot H^2 \log(HT/p)/\epsilon^2$ and union bound on all $T$ iterations. Similarly we can verify (3). For (1), fix model $M \in \mathcal{M}$ and $g \in \mathcal{G}$, by Hoeffding's inequality, with probability at least $1 - p/3$,

$$\left| \mathop{\mathbb{E}}_{(x,a,b)\sim\mathcal{P}^{M_2}_{M_1}} \mathop{\mathbb{E}}_{(r,x)\sim M}[g(x_h, a_h, b_h, r, x) - g(x_h, a_h, b_h, r_h, x_{h+1})] \right. \qquad (19)$$

$$\left. - \sum_{i=1}^{n} \frac{1}{n} \mathop{\mathbb{E}}_{(r,x)\sim M}[g(x_h^i, a_h^i, b_h^i, r, x') - g(x_h^i, a_h^i, b_h^i, r_h^i, x_{h+1}^i)] \right| \qquad (20)$$

$$\leq \sqrt{\frac{\log(2/p)}{n}}. \qquad (21)$$

Thus (1) follows from $n = C \cdot H^2 W_\kappa |\mathcal{A}| \log(T|\mathcal{M}||\mathcal{G}|/p)/(\kappa\epsilon)^2$ and union bound on all $k \in [T]$, $M \in \mathcal{M}$ and $g \in \mathcal{G}$. We complete the proof. $\qquad\square$

**Step 3: $M^*$ stays in the constraint set while it shrinks**   In this step, we show that the constraint set always contains the true environment. The result is displayed in the following.

**Lemma C.8.** *Suppose for each round $k$, the event in Lemma C.7 holds. Then $M^* \in \mathcal{M}^k$ for all $k$. Furthermore, let $\widehat{\mathcal{M}}^k = \{M \in \widehat{\mathcal{M}}^{k-1} : A_{h^k}(M_1^k, M_2^k, M) \leq 2\phi\}$ with $\widehat{\mathcal{M}}^0 = \mathcal{M}$. We have $\mathcal{M}^k \subset \widehat{\mathcal{M}}^k$ for all $k$.*

*Proof.* Since $\mathcal{E}(M_1^k, M_2^k, M^*, h) = 0, \forall h \in [H]$, we have $\widehat{\mathcal{E}}(M_1^k, M_2^k, M^*, h^k) \leq \mathcal{E}(M_1^k, M_2^k, M^*, h^k) + \phi \leq \phi$, Thus $\mathcal{M}^*$ is not eliminated. We prove the second result by induction. Firstly $\widehat{\mathcal{M}}^0 = \mathcal{M}^0$. Assume $\mathcal{M}^{k-1} \subset \widehat{\mathcal{M}}^{k-1}$. For all $M \in \mathcal{M}^k$, we have $A_{h^k}(M_1^k, M_2^k, M) \leq \mathcal{E}(M_1^k, M_2^k, M, h^k) \leq \widehat{\mathcal{E}}(M_1^k, M_2^k, M, h^k) + \phi \leq 2\phi$ by Line 13 of Algorithm 4. Therefore $M \in \widehat{\mathcal{M}}^k$, we complete the proof. $\qquad\square$

**Step 4: Terminate or rank increase**   The following Lemma is key to the 'alternate pessimism principle'. Intuitively, it indicates that the algorithm either terminates and outputs a pair of policies with small duality gap, or detects a time step that provides useful information (large Bellman error).

**Lemma C.9.** *Suppose for round $k$, the event in Lemma C.7 holds and $M^*$ is not eliminated. Then if the algorithm terminate, it outputs a policy pair $\pi^k$ such that $V^*(x_1) - V^{\pi^k, \nu^*_{\pi^k}}(x_1) \leq \epsilon$; otherwise $A_{h^k}((M_1^k, M_2^k), M_i^k) \geq \frac{\kappa\epsilon}{8H}$ for some $i \in [2]$.*

*Proof.* Consider the situation that the algorithm does not terminate. Then with out loss of generality we can assume $|\widehat{V}^k - Q^{M_1^k}(x_1, \pi^k, \nu^k)| \geq \epsilon/2$. By Lemma C.7 and Lemma C.6, we have

$$\left| \sum_{h=1}^{H} \mathcal{L}(M_1^k, M_2^k, M_1^k, h) \right| = |Q_M(x_1, \pi, \nu) - V_1^{\pi, \nu}(x_1)|$$
$$\geq |\widehat{V}^k - Q^{M_1^k}(x_1, \pi^k, \nu^k)| - \left| V^{\pi^k, \nu^k} - \widehat{V}^k \right|$$
$$\geq 3\epsilon/8.$$

By pigeonhole principle, these exists $h \in [H]$ such that $|\mathcal{L}(M_1^k, M_2^k, M_1^k, h)| \geq \frac{3\epsilon}{8H}$. For this $h$ we have $|\widehat{\mathcal{L}}(M_1^k, M_2^k, M_1^k, h)| \geq \frac{3\epsilon}{8H} - \frac{\epsilon}{8H} = \frac{\epsilon}{4H}$, thus the $h^k$ in Line 11 is well defined and we have $|\widehat{\mathcal{L}}(M_1^k, M_2^k, M_1^k, h^k)| \geq \epsilon/(4H)$. Using Lemma C.7 again, we have

$$A_{h^k}((M_1^k, M_2^k), M_i^k) \geq \kappa \cdot |\mathcal{L}(M_1^k, M_2^k, M_1^k, h^k)| \geq \frac{\kappa\epsilon}{8H}.$$

If the algorithm terminates, we have

$$Q^{M_1^k}(x_1, \pi^k, \nu^k) \geq Q^{M_1^k}(x_1, \pi^k, \nu_{\pi^k}^{M_1^k}) \geq V^*(x_1)$$

and $Q^{M_2^k}(x_1, \pi^k, \nu^k) \leq V^{\pi^k, \nu^*_{\pi^k}}(x_1)$. Therefore

$$V^*(x_1) - V^{\pi^k, \nu^*_{\pi^k}}(x_1) \leq Q^{M_1^k}(x_1, \pi^k, \nu^k) - Q^{M_2^k}(x_1, \pi^k, \nu^k)$$
$$\leq |Q^{M_1^k}(x_1, \pi^k, \nu^k) - \widehat{V}^{\pi^k, \nu^k}| + |\widehat{V}^{\pi^k, \nu^k} - Q^{M_2^k}(x_1, \pi^k, \nu^k)|$$
$$\leq \epsilon.$$

We thus complete the proof. $\qquad\square$

**Step 5: Bounding the number of iterations.**   This step uses the technique of Jiang et al. (2017); Sun et al. (2019a), which uses the volume of minimum volume enclosing ellipsoid as a potential function to bound the number of iterations. The key result is presented in Lemma C.10.

**Lemma C.10.** *Suppose for every round $k$, the event in Lemma C.7 holds, then the number of iterations of Algorithm 4 is at most $HW_\kappa \log(\frac{\beta}{2\phi})/\log(5/3)$ for certain $i \in [2]$.*

*Proof.* If the algorithm does not terminate at round $k$, then by Lemma C.9, $\langle \zeta_{h^k}(M_1^k, M_2^k), \chi_{h^k}(M_i^k) \rangle = A_{h^k}((M_1^k, M_2^k), M_i^k) \geq \frac{\kappa\epsilon}{8H} = 6\sqrt{W_\kappa}\phi$. Let $O_h^k$ denote the origin-centered minimum volume enclosing ellipsoid of $\{\chi_h(M) : M \in \widehat{\mathcal{M}}_k\}$. Let $O_h^{k-1,+}$ denote the origin-centered minimum volume enclosing ellipsoid of $\{v \in O_h^{k-1} : \langle \zeta_{h^k}(M_1^k, M_2^k), v \rangle \leq 2\phi\}$. By Lemma F.2, we have

$$\frac{\text{vol}(O_{h^k}^k)}{\text{vol}(O_{h^k}^{k-1})} \leq \frac{\text{vol}(O_{h^k}^{k-1,+})}{\text{vol}(O_{h^k}^{k-1})} \leq 3/5.$$

Define $\Phi = \sup_{M_1, M_2 \in \mathcal{M}, h \in [H]} \|\zeta_h(M_1, M_2)\|_2$ and $\Psi = \sup_{M \in \mathcal{M}, h \in [H]} \|\chi_h(M)\|_2$. Then we have $\text{vol}(O_h^0) \leq V_{W_\kappa}(1) \cdot \Psi^{W_\kappa}$ where $V_{W_\kappa}(R)$ is the volume of Euclidean ball in $\mathbb{R}^{W_\kappa}$ with radius $R$. On the other hand, we have

$$\text{vol}(O_h^k) \geq \text{vol}(\{q \in \mathbb{R}^{W_\kappa} : \|q\|_2 \leq 2\phi/\Phi\}) \geq V_{W_\kappa}(1) \cdot (2\phi/\Phi)^{W_\kappa}.$$

It thus follows that the number of iterations for each $h$ is at most $W_\kappa \log(\frac{\Phi\Psi}{2\phi})/\log(5/3)$. Using $\beta \geq \Phi\Psi$, this completes the proof. $\qquad\square$

**Step 6: Putting everything together.** From Lemma C.7, with high probability the events holds. Therefore for the first $T$ rounds, we can apply Lemma C.10 and know that the algorithm indeed terminates in at most $T$ rounds. The number of trajectories is thus upper bounded by $(n_1 + n)T = O(\frac{H^3 W_\kappa^2 |\mathcal{A}|}{\kappa^2 \epsilon^2} \log(T|\mathcal{G}||\mathcal{M}|/p))$.

## D  THEORY FOR ALTERNATE OPTIMISTIC VALUE ELIMINATION (AOVE)

We prove Theorem 4.4 in the following five steps. In the first step we show that the functions in the constraint set has low Bellman error. Notice that different from Lemma B.1, here we consider a different Bellman operator in Eq (2) and consider all policies in policy class $\Pi$. In the second step we show that $Q^{\pi, \nu_\pi^*}$ belongs to the constraint set for all $\pi \in \Pi$ and in particular $Q^*$ belongs to the constraint set. In the third step we perform a regret decomposition. Notice that here we consider two Bellman errors. We then bound the summation of these Bellman errors by Minimax Eluder dimension in step 4 and complete the proof in step 5.

**Step 1: Low Bellman errors in the constraint set.**

**Lemma D.1** (Concentration on Bellman error). *Let $\rho > 0$ be an arbitrary fixed number. With probability at least $1 - p$ for all $(k, h) \in [K] \times [H]$ we have*

$$(a) \sum_{i=1}^{k-1} \left( f_h^k(x_h^i, a_h^i, b_h^i) - r_h^i - \mathop{\mathbb{E}}_{x \sim \mathbb{P}(\cdot|x_h^i, a_h^i, b_h^i)} f_{h+1}^k(x, \pi_{h+1}^k, (\nu_{\pi^k}^{f^k})_{h+1})) \right)^2 \leq O(\beta)$$

$$(b) \sum_{i=1}^{k-1} \left( g_h^k(x_h^i, a_h^i, b_h^i) - r_h^i - \mathop{\mathbb{E}}_{x \sim \mathbb{P}(\cdot|x_h^i, a_h^i, b_h^i)} g_{h+1}^k(x, \pi_{h+1}^k, (\nu_{\pi^k}^{g^k})_{h+1})) \right)^2 \leq O(\beta)$$

*Proof.* For simplicity we only proof $(a)$, and $(b)$ follows similarly.

Consider a fixed $(k, h, f, \pi)$ tuple, let

$$U_t(h, f, \pi) := \left( f_h(x_h^t, a_h^t, b_h^t) - r_h^t - f_{h+1}(x_{h+1}^t, \pi, \nu_\pi^f) \right)^2$$
$$- \left( r_h^t + \mathop{\mathbb{E}}_{x \sim \mathbb{P}(\cdot|x_h^t, a_h^t, b_h^t)} f_{h+1}(x, \pi, \nu_\pi^f) - r_h^t - f_{h+1}(x_{h+1}^t, \pi, \nu_\pi^f) \right)^2$$

and $\mathscr{F}_{t,h}$ be the filtration induced by $\{x_1^i, a_1^i, b_1^i, r_1^i, \ldots, x_H^i\}_{i=1}^{t-1} \cup \{x_1^t, a_1^t, b_1^t, r_1^t, \ldots, x_h^t, a_h^t, b_h^t\}$.

We have

$$\mathbb{E}[U_t(h, f, \pi)|\mathscr{F}_{t,h}]$$
$$= [(f_h(x_h^t, a_h^t, b_h^t) - r_h^t - \underset{x \sim \mathbb{P}(\cdot|\cdot,\cdot,\cdot)}{\mathbb{E}} f_{h+1}(x, \pi, \nu_\pi^f))(x_h^t, a_h^t, b_h^t)]$$
$$\cdot \mathbb{E}[f_h(x_h^t, a_h^t, b_h^t) + r_h^t + \underset{x \sim \mathbb{P}(\cdot|x_h^t, a_h^t, b_h^t)}{\mathbb{E}} f_{h+1}(x, \pi, \nu_\pi^f) - 2r_h^t - 2f_{h+1}(x_{h+1}^t, \pi, \nu_\pi^f)|\mathscr{F}_{t,h}]$$
$$= [f_h(x_h^t, a_h^t, b_h^t) - r_h^t - \underset{x \sim \mathbb{P}(\cdot|x_h^t, a_h^t, b_h^t)}{\mathbb{E}} f_{h+1}(x, \pi, \nu_\pi^f)]^2$$

and

$$\mathrm{Var}[U_t(h, f, \pi)|\mathscr{F}_{t,h}] \le \mathbb{E}[(U_t(h, f, \pi))^2|\mathscr{F}_{t,h}]$$
$$= [(f_h - r_h - \underset{x \sim \mathbb{P}(\cdot|\cdot,\cdot,\cdot)}{\mathbb{E}} f_{h+1}(x, \pi, \nu_\pi^f))(x_h^t, a_h^t, b_h^t)]^2$$
$$\cdot \mathbb{E}[(f_h(x_h^t, a_h^t, b_h^t) + r_h^t + \underset{x \sim \mathbb{P}(\cdot|x_h^t, a_h^t, b_h^t)}{\mathbb{E}} f_{h+1}(x, \pi, \nu_\pi^f) - 2r_h^t - 2f_{h+1}(x_{h+1}^t, \pi, \nu_\pi^f))^2|\mathscr{F}_{t,h}]$$
$$\le 36[f_h(x_h^t, a_h^t, b_h^t) - r_h^t - \underset{x \sim \mathbb{P}(\cdot|x_h^t, a_h^t, b_h^t)}{\mathbb{E}} f_{h+1}(x, \pi, \nu_\pi^f)]^2 = 36\,\mathbb{E}[U_t(h, f, \pi)|\mathscr{F}_{t,h}].$$

By Freedman's inequality, we have with probability at least $1 - p$,

$$\left| \sum_{t=1}^k U_t(h, f, \pi) - \sum_{t=1}^k \mathbb{E}[U_t(h, f, \pi)|\mathscr{F}_{t,h}] \right| \le O\left( \sqrt{\log(1/p) \sum_{t=1}^k \mathbb{E}[U_t|\mathscr{F}_{t,h}]} + \log(1/p) \right).$$

Let $\mathscr{N}_\rho \times \mathscr{Y}_\rho$ be a $\rho$-cover of $\mathcal{F} \times \Pi$. The covering of $\Pi$ is in-terms of the distance of two policies defined as

$$d(\pi, \pi') := \max_{f \in \mathcal{F}, x \in \mathcal{S}, b \in \mathcal{A}} |f(x, \pi, b) - f(x, \pi', b)|.$$

Taking a union bound for all $(k, h, \mathfrak{f}, \mathfrak{p}) \in [K] \times [H] \times \mathscr{N}_\rho \times \mathscr{Y}_\rho$, we have that with probability at least $1 - p$ for all $(k, h, \mathfrak{f}, \mathfrak{p}) \in [K] \times [H] \times \mathscr{N}_\rho \times \mathscr{Y}_\rho$

$$\left| \sum_{t=1}^k U_t(h, \mathfrak{f}, \mathfrak{p}) - \sum_{t=1}^k [\mathfrak{f}_h(x_h^t, a_h^t, b_h^t) - r_h^t - \underset{x \sim \mathbb{P}(\cdot|x_h^t, a_h^t, b_h^t)}{\mathbb{E}} \mathfrak{f}_{h+1}(x, \mathfrak{p}, \nu_\mathfrak{p}^\mathfrak{f})]^2 \right|$$
$$\le O\left( \sqrt{\iota \sum_{t=1}^k [\mathfrak{f}_h(x_h^t, a_h^t, b_h^t) - r_h^t - \underset{x \sim \mathbb{P}(\cdot|x_h^t, a_h^t, b_h^t)}{\mathbb{E}} \mathfrak{f}_{h+1}(x, \mathfrak{p}, \nu_\mathfrak{p}^\mathfrak{f})]^2} + \iota \right) \tag{22}$$

where $\iota = \log(HK|\mathscr{N}_\rho||\mathscr{Y}_\rho|/p)$. For an arbitrary $(h, k) \in [H] \times [K]$ pair, by the definition of $\mathcal{V}^k$ and $(\pi^k, f^k) \in \mathcal{V}^k$ we have

$$\sum_{t=1}^k U_t(h, f^k, \pi^k)$$
$$- \sum_{t=1}^k \left( r_h^t + \underset{x \sim \mathbb{P}(\cdot|x_h^t, a_h^t, b_h^t)}{\mathbb{E}} f_{h+1}^k(x, \pi^k, \nu_{\pi^k}^{f^k}) - r_h^t - f_{h+1}^k(x_{h+1}^t, \pi^k, \nu_{\pi^k}^{f^k}) \right)^2$$
$$\le \sum_{t=1}^k \left( f_h^k(x_h^t, a_h^t, b_h^t) - r_h^t - f_{h+1}^k(x_{h+1}^t, \pi^k, \nu_{\pi^k}^{f^k}) \right)^2$$
$$- \inf_{g \in \mathcal{F}} \sum_{t=1}^k \left( g_h^k(x_h^t, a_h^t, b_h^t) - r_h^t - f_{h+1}^k(x_{h+1}^t, \pi^k, \nu_{\pi^k}^{f^k}) \right)^2$$
$$\le O(\iota + k\rho) \tag{23}$$

where in the first step we use Assumption 3.2.

Define $\mathfrak{f}^k = \operatorname{argmin}_{\mathfrak{f} \in \mathcal{N}_\rho} \max_{h \in [H]} \|f_h^k - \mathfrak{f}_h^k\|_\infty, \mathfrak{p}^k = \operatorname{argmin}_{\mathfrak{p} \in \mathcal{Y}_\rho} d(\pi^k, \mathfrak{p})$. Since $\mathcal{N}_\rho$ is a $\rho$-cover of $\mathcal{F}$,

$$\left| \sum_{t=1}^k U_t(h, f^k, \pi^k) - \sum_{t=1}^k U_t(h, \mathfrak{f}^k, \mathfrak{p}^k) \right| \le O(k\rho). \tag{24}$$

Therefore combining Eq (23) and Eq (24)

$$\sum_{t=1}^k U_t(h, \mathfrak{f}^k, \mathfrak{p}^k) \le O(\iota + k\rho). \tag{25}$$

Recall Eq (22) indicates

$$\left| \sum_{t=1}^k U_t(h, \mathfrak{f}^k, \mathfrak{p}^k) - \sum_{t=1}^k [\mathfrak{f}_h^k(x_h^t, a_h^t, b_h^t) - r_h^t - \mathbb{E}_{x \sim \mathbb{P}(\cdot | x_h^t, a_h^t, b_h^t)} \mathfrak{f}_{h+1}^k(x, \mathfrak{p}^k, \nu_{\mathfrak{p}^k}^{\mathfrak{f}^k})]^2 \right|$$

$$\le O\left( \sqrt{\iota \sum_{t=1}^k [\mathfrak{f}_h^k(x_h^t, a_h^t, b_h^t) - r_h^t - \mathbb{E}_{x \sim \mathbb{P}(\cdot | x_h^t, a_h^t, b_h^t)} \mathfrak{f}_{h+1}^k(x, \mathfrak{p}^k, \nu_{\mathfrak{p}^k}^{\mathfrak{f}^k})]^2} + \iota \right) \tag{26}$$

Combining Eq (25) and Eq (26) we have

$$\sum_{t=1}^k [\mathfrak{f}_h^k(x_h^t, a_h^t, b_h^t) - r_h^t - \mathbb{E}_{x \sim \mathbb{P}(\cdot | x_h^t, a_h^t, b_h^t)} \mathfrak{f}_{h+1}^k(x, \mathfrak{p}^k, \nu_{\mathfrak{p}^k}^{\mathfrak{f}^k})]^2 \le O(\iota + k\rho).$$

Since $\mathfrak{f}^k$ is an $\rho$-approximation to $f^k$ and $\mathfrak{p}^k$ is an $\rho$-approximation to $\pi^k$, we can conclude the proof of $(a)$ with

$$\sum_{i=1}^{k-1} \left( f_h^k(x_h^i, a_h^i, b_h^i) - r_h^i - \mathbb{E}_{x \sim \mathbb{P}(\cdot | x_h^i, a_h^i, b_h^i)} f_{h+1}^k(x, \pi_{h+1}^k, (\nu_{\pi^k}^{f^k})_{h+1})) \right)^2 \le O(\beta).$$

$\square$

**Step 2:** $(\pi, Q^{\pi, \nu_\pi^*}), \forall \pi \in \Pi$ **is always in constraint set.**

**Lemma D.2.** *With probability at least $1 - p$, we have $(\pi^*, Q^*)$ and $(\pi, Q^{\pi, *}), \forall \pi \in \Pi$ all belong to $\mathcal{V}^k$ for all $k \in [K]$.*

*Proof.* Consider a fixed $(k, h, f, \pi)$ tuple, let

$$U_t(h, f, \pi) := \left( f_h(x_h^t, a_h^t, b_h^t) - r_h^t - Q_{h+1}^{\pi, \nu_\pi^*}(x_{h+1}^t, \pi, \nu_\pi^*) \right)^2$$
$$- \left( Q_h^{\pi, \nu_\pi^*}(x_h^t, a_h^t, b_h^t) - r_h^t - Q_{h+1}^{\pi, \nu_\pi^*}(x_{h+1}^t, \pi, \nu_\pi^*) \right)^2$$

and $\mathscr{F}_{t,h}$ be the filtration induced by $\{x_1^i, a_1^i, b_1^i, r_1^i, \ldots, x_H^i\}_{i=1}^{t-1} \cup \{x_1^t, a_1^t, b_1^t, r_1^t, \ldots, x_h^t, a_h^t, b_h^t\}$. We have

$$\mathbb{E}[U_t(h, f, \pi) | \mathscr{F}_{t,h}] = [(f_h - Q_h^{\pi, \nu_\pi^*})(x_h^t, a_h^t, b_h^t)]^2$$

and

$$\mathrm{Var}[U_t(h, f, \pi) | \mathscr{F}_{t,h}] \le 36[(f_h - Q_h^{\pi, \nu_\pi^*})(x_h^t, a_h^t, b_h^t)]^2 = 36 \mathbb{E}[U_t(h, f, \pi) | \mathscr{F}_{t,h}].$$

By Freedman's inequality, we have with probability at least $1 - p$,

$$\left| \sum_{t=1}^k U_t(h, f, \pi) - \sum_{t=1}^k [(f_h - Q_h^{\pi, \nu_\pi^*})(x_h^t, a_h^t, b_h^t)]^2 \right|$$

$$\le O\left( \sqrt{\log(1/p) \sum_{t=1}^k [(f_h - Q_h^{\pi, \nu_\pi^*})(x_h^t, a_h^t, b_h^t)]^2} + \log(1/p) \right).$$

Let $\mathcal{N}_\rho \times \mathcal{Y}_\rho$ be a $\rho$-cover of $\mathcal{F} \times \Pi$. Taking a union bound for al $(k, h, \mathfrak{f}, \mathfrak{p}) \in [K] \times [H] \times \mathcal{N}_\rho \times \mathcal{Y}_\rho$, we have that with probability at least $1 - p$ for all $(k, h, \mathfrak{f}, \mathfrak{p}) \in [K] \times [H] \times \mathcal{N}_\rho \times \mathcal{Y}_\rho$

$$- \sum_{t=1}^{k} U_t(h, \mathfrak{f}, \mathfrak{p}) \leq O(\iota)$$

where $\iota = \log(HK|\mathcal{N}_\rho|/p)$. This implies for all $(k, h, \mathfrak{f}, \mathfrak{p}) \in [K] \times [H] \times \mathcal{N}_\rho \times \mathcal{Y}_\rho$

$$\sum_{t=1}^{k} \left(Q_h^{\mathfrak{p}, \nu_\mathfrak{p}^*}(x_h^t, a_h^t, b_h^t) - r_h^t - Q_{h+1}^{\mathfrak{p}, \nu_\mathfrak{p}^*}(x_{h+1}^t, \mathfrak{p}, \nu_\mathfrak{p}^*)\right)^2$$

$$\leq \sum_{t=1}^{k} \left(\mathfrak{f}_h(x_h^t, a_h^t, b_h^t) - r_h^t - Q_{h+1}^{\mathfrak{p}, \nu_\mathfrak{p}^*}(x_{h+1}^t, \mathfrak{p}, \nu_\mathfrak{p}^*)\right)^2 + O(\iota + k\rho).$$

Since $\mathcal{N}_\rho \times \mathcal{Y}_\rho$ be a $\rho$-cover of $\mathcal{F} \times \Pi$, for all $\mathfrak{p}^k \in \mathcal{Y}_\rho$

$$\sum_{t=1}^{k} \left(Q_h^{\mathfrak{p}, \nu_\mathfrak{p}^*}(x_h^t, a_h^t, b_h^t) - r_h^t - Q_{h+1}^{\mathfrak{p}, \nu_\mathfrak{p}^*}(x_{h+1}^t, \mathfrak{p}, \nu_\mathfrak{p}^*)\right)^2$$

$$\leq \inf_{g \in \mathcal{F}} \sum_{t=1}^{k} \left(g_h(x_h^t, a_h^t, b_h^t) - r_h^t - Q_{h+1}^{\mathfrak{p}, \nu_\mathfrak{p}^*}(x_{h+1}^t, \mathfrak{p}, \nu_\mathfrak{p}^*)\right)^2 + O(\iota + k\rho).$$

Again using $\mathcal{N}_\rho \times \mathcal{Y}_\rho$ be a $\rho$-cover of $\mathcal{F} \times \Pi$, we have for all $\pi \in \Pi$

$$\sum_{t=1}^{k} \left(Q_h^{\pi, \nu_\pi^*}(x_h^t, a_h^t, b_h^t) - r_h^t - Q_{h+1}^{\pi, \nu_\pi^*}(x_{h+1}^t, \pi, \nu_\pi^*)\right)^2$$

$$\leq \inf_{g \in \mathcal{F}} \sum_{t=1}^{k} \left(g_h(x_h^t, a_h^t, b_h^t) - r_h^t - Q_{h+1}^{\pi, \nu_\pi^*}(x_{h+1}^t, \pi, \nu_\pi^*)\right)^2 + O(\iota + k\rho).$$

Therefore we have $(\pi, Q^{\pi, \nu_\pi^*}), \forall \pi \in \Pi$ all belong to $\mathcal{V}^k$ for all $k \in [K]$ with probability at least $1 - p$. $\qquad\square$

**Remark D.3.** *An important implication of this fact is the following*

$$V^* - V^{\pi^k, \nu_{\pi^k}^*} \leq f_1^k(x_1^k, \pi^k, \nu_{\pi^k}^{f^k}) - g_1^k(x_1^k, \pi^k, \nu^k)$$

*which gives an upper bound of the one-side duality gap.*

**Step 3: Bounding the regret by Bellman error and martingale.**

**Lemma D.4.** *Define*

$$\delta_h^k := f_h^k(x_h^k, \pi_h^k, (\nu_{\pi^k}^{f^k})_h) - g_h^k(x_h^k, \pi_h^k, \nu_h^k)$$

$$\zeta_h^k := \mathbb{E}[\delta_{h+1}^k | x_h^k, a_h^k, b_h^k] - \delta_{h+1}^k$$

$$\gamma_h^k := \mathbb{E}_{a \sim \pi_h^k, b \sim \nu_h^k}[f_h^k(x_h^k, a, b)] - f_h^k(x_h^k, a_h^k, b_h^k)$$

$$\widehat{\gamma}_h^k := \mathbb{E}_{a \sim \pi_h^k, b \sim \nu_h^k}[g_h^k(x_h^k, a, b)] - g_h^k(x_h^k, a_h^k, b_h^k)$$

*Then we have*

$$\delta_h^k \leq \delta_{h+1}^k + \zeta_h^k + \epsilon_h^k - \widehat{\epsilon}_h^k + \gamma_h^k - \widehat{\gamma}_h^k$$

*where $\epsilon_h^k, \widehat{\epsilon}_h^k$ are the bellman error defined by*

$$\epsilon_h^k := f_h^k(x_h^k, a_h^k, b_h^k) - r_h^k - \mathbb{E}_{x \sim \mathbb{P}(\cdot | x_h^k, a_h^k, b_h^k)} f_h^k(x, \pi_{h+1}^k, (\nu_{\pi^k}^{f^k})_{h+1})$$

$$\widehat{\epsilon}_h^k := g_h^k(x_h^k, a_h^k, b_h^k) - r_h^k - \mathbb{E}_{x \sim \mathbb{P}(\cdot | x_h^k, a_h^k, b_h^k)} g_h^k(x, \pi_{h+1}^k, \nu_{h+1}^k).$$

*Proof.* By definition of $(\nu_{\pi^k}^{f^k})_h = \operatorname{argmin}_{\nu \in \Pi_h} f_h^k(x_h^k, \pi_h^k, \nu)$, we have

$$f_h^k(x_h^k, \pi_h^k, (\nu_{\pi^k}^{f^k})_h) \leq f_h^k(x_h^k, \pi_h^k, \nu_h^k)$$
$$= f_h^k(x_h^k, a_h^k, b_h^k) + \gamma_h^k$$

It thus follows that

$$
\begin{aligned}
\delta_h^k &= f_h^k(x_h^k, \pi_h^k, (\nu_{\pi^k}^{f^k})_h) - g_h^k(x_h^k, \pi_h^k, \nu_h^k) \\
&\leq f_h^k(x_h^k, a_h^k, b_h^k) - g_h^k(x_h^k, a_h^k, b_h^k) + \gamma_h^k - \widehat{\gamma}_h^k \\
&= \Big( r_h^k + \mathop{\mathbb{E}}_{x \sim \mathbb{P}(\cdot | x_h^k, a_h^k, b_h^k)} f_h^k(x, \pi_{h+1}^k, (\nu_{\pi^k}^{f^k})_{h+1}) + \epsilon_h^k \Big) \\
&\quad - \Big( r_h^k + \mathop{\mathbb{E}}_{x \sim \mathbb{P}(\cdot | x_h^k, a_h^k, b_h^k)} g_h^k(x, \pi_{h+1}^k, \nu_{h+1}^k) + \widehat{\epsilon}_h^k \Big) + \gamma_h^k - \widehat{\gamma}_h^k \\
&= \mathop{\mathbb{E}}_{x \sim \mathbb{P}(\cdot | x_h^k, a_h^k, b_h^k)} f_h^k(x, \pi_{h+1}^k, (\nu_{\pi^k}^{f^k})_{h+1}) - \mathop{\mathbb{E}}_{x \sim \mathbb{P}(\cdot | x_h^k, a_h^k, b_h^k)} g_h^k(x, \pi_{h+1}^k, \nu_{h+1}^k) \\
&\quad + \epsilon_h^k - \widehat{\epsilon}_h^k + \gamma_h^k - \widehat{\gamma}_h^k \\
&= \delta_{h+1}^k + \zeta_h^k + \epsilon_h^k - \widehat{\epsilon}_h^k + \gamma_h^k - \widehat{\gamma}_h^k.
\end{aligned}
$$

Thus we complete the proof. □

### Step 4: Bounding cumulative Bellman error using Minimax Eluder dimension.
**Lemma D.5** (Bounding Bellman error). *For any $(k, h) \in [K] \times [H]$ we have*

$$\sum_{t=1}^{k} |\epsilon_h^t| \leq O\bigg( \sqrt{k \cdot \dim_{\mathrm{ME}'}(\mathcal{F}, \sqrt{1/K}) \log[\mathcal{N}(\mathcal{F}, 1/K) \cdot \mathcal{N}(\Pi, 1/K) \cdot HK/p]} \bigg)$$

$$\sum_{t=1}^{k} |\widehat{\epsilon}_h^t| \leq O\bigg( \sqrt{k \cdot \dim_{\mathrm{ME}'}(\mathcal{F}, \sqrt{1/K}) \log[\mathcal{N}(\mathcal{F}, 1/K) \cdot \mathcal{N}(\Pi, 1/K) \cdot HK/p]} \bigg).$$

*Proof.* Let

$$l_t(\cdot, \cdot, \cdot) = f_h^t(\cdot, \cdot, \cdot) - r_h(\cdot, \cdot, \cdot) - \mathop{\mathbb{E}}_{x \sim \mathbb{P}(\cdot, \cdot, \cdot)} f_{h+1}^t(x, \pi^t, \nu_{\pi^t}^{f^t})$$

and $\rho_t = \delta(x_h^t, a_h^t, b_h^t)$. Similarly, we can also let $\rho_t$ be generated by $(\pi^t, \nu^t)$. Then Lemma D.1 indicates

$$\sum_{t=1}^{k} l_k^2(\rho_t) \leq O(\beta).$$

Based on this, we can apply Lemma F.1 to obtain the first inequality. The second follows similarly. □

### Step 5: Putting them all together.
We can upper bound the regret as follows:

$$
\begin{aligned}
\sum_{k=1}^{K} [V^* - V^{\pi^k, \nu_{\pi^k}^*}] &\leq \sum_{k=1}^{K} [f_1^k(x_1^k, \pi^k, \nu_{\pi^k}^{f^k}) - g_1^k(x_1^k, \pi^k, \nu^k)] \\
&\leq \sum_{k=1}^{K} \sum_{h=1}^{H} (\zeta_h^k + \epsilon_h^k - \widehat{\epsilon}_h^k + \gamma_h^k - \widehat{\gamma}_h^k) \\
&= \sum_{h=1}^{H} \sum_{k=1}^{K} (\zeta_h^k + \gamma_h^k - \widehat{\gamma}_h^k) + \sum_{h=1}^{H} \sum_{k=1}^{K} (\epsilon_h^k - \widehat{\epsilon}_h^k) \qquad (27)
\end{aligned}
$$

where the first step comes from Lemma D.2, the second step comes from Lemma D.4 and the last step comes from Fubini's theorem. Notice that the first term in Eq (27) is a martingale difference sequence and can be bounded by $O(H\sqrt{KH \log(HK/p)})$ with probability at least $1 - p$ and the second term can be bounded by Lemma D.5. We thus complete the proof.

# E    EXAMPLES

In this section we give several examples of Markov games that possess low Minimax Eluder dimension, including *tabular Markov Games, linear function approximation, reproducing kernel Hilbert space (RKHS) function approximation, overparameterized neural networks, and generalized linear models*. We first consider reproducing kernel Hilbert space (RKHS) function approximation, which captures many existing models.

**Reproducing kernel Hilbert space (RKHS) function approximation.** In this setting, the function class of interest is given by

$$\mathcal{F}_h = \{\langle \phi_h(x, a, b), \theta_h \rangle : \theta, \phi(\cdot, \cdot, \cdot) \in B_R(\mathcal{H})\}.$$

Here $\phi(\cdot, \cdot, \cdot) : \mathcal{S} \times \mathcal{A} \times \mathcal{A} \mapsto \mathcal{H}$ is a feature map to $\mathcal{H}$, $\theta \in \mathcal{H}$, and $B_R(\mathcal{H})$ is the ball centered at zero with radius $R$ in a Hilbert space $\mathcal{H}$. The effective dimension of a set $\mathcal{D} \subset \mathcal{H}$ is characterized by the critical information gain $\widetilde{d}(\gamma, \mathcal{D})$. Specifically, for $\gamma > 0$ and $\mathcal{D} \subset \mathcal{H}$, we define $d_n(\gamma, \mathcal{D}) = \max_{x_1, \dots, x_n \in \mathcal{D}} \log \det(I + \frac{1}{\gamma} \sum_{i=1}^n x_i x_i^\top)$. The critical information gain (Du et al., 2021) is thus defined by $\widetilde{d}(\gamma, \mathcal{D}) = \min_{k \in \mathbb{N} : k \geq d_n(\gamma, \mathcal{D})} k$. The main result is given in the following theorem.

**Theorem E.1.** *In kernel function approximation, the Minimax Eluder dimensions* $\dim_{\mathrm{ME}}(\mathcal{F}, \epsilon)$ *and* $\dim_{\mathrm{ME}'}(\mathcal{F}, \epsilon)$ *are upper bounded by* $\max_{h \in [H]} \widetilde{\gamma}(\frac{\epsilon}{9R}, \Phi_h)$, *where* $\Phi_h = \{\phi_h(x, a, b) : x \in \mathcal{S}, a, b \in \mathcal{A}\}$.

*Proof.* We only show $\dim_{\mathrm{ME}'}(\mathcal{F}, \epsilon) \leq \max_{h \in [H]} \widetilde{\gamma}(\frac{\epsilon}{9R}, \Phi_h)$, and the other part is totally similar. Fix an $h \in [H]$. Due to completeness, for any $\pi \in \Pi$ and $f \in \mathcal{F}$ we have $(f_h - \mathcal{T}_h^\pi f_{h+1})(x, a, b) = \langle \phi_h(x, a, b), \theta_h - \mathcal{T}_h^\pi \theta_{h+1} \rangle$ where $\mathcal{T}_h^\pi \theta_{h+1}$ is the unique feature in $\mathcal{H}$ such that $\mathcal{T}_h^\pi f_{h+1} = \langle \phi_h(x, a, b), \mathcal{T}_h^\pi \theta_{h+1} \rangle$. Consider the longest sequence $\nu_1, \dots, \nu_k \in \mathcal{D}_\Delta$ such that each one item is $\epsilon$-independent of its precedents, distinguished by a sequence of features $\theta_h^{(1)}, \dots, \theta_h^{(k)}$. To simplify notations, we define $\phi_i = \mathbb{E}_{\nu_i}[\phi_h(x, a, b)]$ and $\theta_i = \theta_h^{(i)} - \mathcal{T}_h^\pi \theta_{h+1}^{(i)}$ for all $i \in [k]$. Let $\Lambda_i = \sum_{\tau=1}^{i-1} \phi_\tau \phi_\tau^\top + \frac{\epsilon^2}{9R^2} I$. Then Definition 3.3 gives

$$\|\phi_i\|_{\Lambda_i^{-1}} \geq \frac{1}{\epsilon} \cdot \|\phi_i\|_{\Lambda_i^{-1}} \cdot \|\theta_i\|_{\Lambda_i} \geq \frac{1}{\epsilon} \cdot |\langle \phi_i, \theta_i \rangle| \geq 1.$$

where the first step comes from $\theta_i \in B_R(\mathcal{H})$ and $\sum_{j=1}^{i-1} \langle \phi_j, \theta_j \rangle^2 \leq \epsilon$, the second step comes from Hölder's inequality and the final step comes from $|\langle \phi_i, \theta_i \rangle| \geq \epsilon$. Thus we have

$$k \leq \sum_{i=1}^k \log(1 + \|\phi_i\|_{\Lambda_i^{-1}}) = \log \det(I + \frac{9R^2}{\epsilon^2} \sum_{i=1}^k \phi_i \phi_i^\top).$$

By the definition of critical information gain, $k$ must be less or equal than $\widetilde{\gamma}(\frac{\epsilon}{9R}, \Phi_h)$.   $\square$

This result implies several important cases in general function approximation.

- **Linear function approximation.**    In this setting $r_h(x, a, b) = \phi(x, a, b)^\top \theta_h$ and $\mathbb{P}_h(\cdot | x, a, b) = \phi(x, a, b)^\top \mu_h(\cdot)$ where $\phi(x, a, b) \in \mathbb{R}^d$. We have shown that this setting satisfies all the realizability and completeness assumptions in the previous sections. By standard Ellipsoid potential arguments, the critical information gain is upper bounded by $O(d \cdot \log(R/\epsilon))$. Therefore, this setting also has low Minimax Eluder dimensions.

- **Tabular Markov Games.**    This setting can be seen as a special case of linear function approximation where the feature vectors are chosen to be one-hot representations of state-actions pairs. In this case, $d = |\mathcal{S}| \cdot |\mathcal{A}|^2$ and thus Minimax Eluder dimensions is upper bounded by $\widetilde{O}(|\mathcal{S}| \cdot |\mathcal{A}|^2)$.

- **Overparameterized neural networks.** In overparameterized neural network function approximations, the functions are given by $h(\phi(x, a, b))$ where $h$ is the Neural Tangent Random Feature space defined in Cao & Gu (2019). We rewrite the information gain $d_n(\gamma, \mathcal{D})$ as $\max_{x_1, \dots, x_n \in \mathcal{D}} \log \det(I + \frac{1}{\gamma} K_n)$ where $(K_n)_{i,j} = K(x_i, x_j)$ and $K$ is the Neural Tangent Kernel. It has been shown (Jacot et al., 2018; Du et al., 2019c;a; Allen-Zhu

et al., 2019) that with proper initialization and learning rates, the function approximations approximately lie in the Reproducing kernel Hilbert space given by the neural tangent feature as above. Therefore the Minimax Eluder dimensions are bounded by $\widetilde{\gamma}(\frac{\epsilon}{9R}, \Phi_h) + \lambda$, where $\widetilde{\gamma}$ is the critical information gain defined by the $d_n(\gamma, \mathcal{D})$ rewritten above and $\lambda$ is a term that vanishes as width tends to infinity.

**Generalized linear models.** Finally, we consider generalized linear model as an extension of the linear function approximations.

**Theorem E.2.** *Suppose* $(f_h - \mathcal{T}_h^\pi f_{h+1})(x, a, b) = g(\phi(x, a, b)^\top \theta_h)$ *for any* $\pi \in \Pi$ *and* $f \in \mathcal{F}$, *where* $g$ *is a differentiable and strictly increasing function and* $\phi, \theta_h \in \mathbb{R}^d$. *Assume* $g' \in (c_1, c_2)$ *and* $\|\phi\|_2, \|\theta_h\|_2 \le R$ *for* $c_1, c_2, R > 0$. *Then* $\dim_{\mathrm{ME}'}(\mathcal{F}, \epsilon) \le O(d(c_2/c_1)^2 \log(\frac{c_2 R}{c_1 \epsilon}))$.

*Proof.* The proof mirrors those in Theorem E.1. For the longest sequence $\nu_1, \ldots, \nu_k \in \mathcal{D}_\Delta$ such that each one item is $\epsilon$-independent of its precedents, distinguished by a sequence of features $\theta_h^{(1)}, \ldots, \theta_h^{(k)}$, we define $\phi_i = \mathbb{E}_{\nu_i}[\phi_h(x, a, b)]$ and $\theta_i = \theta_h^{(i)} - \mathcal{T}_h^\pi \theta_{h+1}^{(i)}$ for all $i \in [k]$. Let $\Lambda_i = \sum_{\tau=1}^{i-1} \phi_\tau \phi_\tau^\top + \frac{\epsilon^2}{9R^2 c_1^2} I$. Then Definition 3.3 gives

$$\|\phi_i\|_{\Lambda_i^{-1}} \ge \frac{c_1}{\epsilon} \cdot \|\phi_i\|_{\Lambda_i^{-1}} \cdot \|\theta_i\|_{\Lambda_i} \ge \frac{c_1}{\epsilon} \cdot |\langle \phi_i, \theta_i \rangle| \ge \frac{c_1}{c_2}.$$

This means $\det(\Lambda_k) \ge (1 + \frac{c_1^2}{c_2^2})^{k-1}$. But by Elliptic potential lemma, $\det(\Lambda_k) \le (\frac{R^2 k}{d} + \frac{\epsilon^2}{9R^2 c_1^2})^d$. Combining these two inequalities gives $k \le O(d(c_2/c_1)^2 \log(\frac{c_2 R}{c_1 \epsilon}))$. $\qquad\square$

## F TECHNICAL CLAIMS

This section lists the facts we use from the previous work.

**Lemma F.1** (Cumulative error control, Lemma 26 of Jin et al. (2021a)). *Given a function class* $\mathcal{G}$ *defined on* $\mathcal{X}$ *with* $|g(x)| \le C$ *for all* $(g, x) \in \mathcal{G} \times \mathcal{X}$ *and a family of probability measures* $\mathcal{D}$ *over* $\mathcal{X}$. *Suppose sequence* $\{g_k\}_{k=1}^K \subset \mathcal{G}$ *and* $\{\rho_k\}_{k=1}^H \subset \mathcal{D}$ *such that for all* $k \in [K]$, $\sum_{t=1}^{k-1} (\mathbb{E}_{\rho_t}[g_k])^2 \le \beta$. *Then for all* $k \in [K]$ *and* $\omega > 0$,

$$\sum_{t=1}^k |\mathbb{E}_{\rho_t}[g_t]| \le O\left( \sqrt{\dim_{\mathrm{DE}}(\mathcal{G}, \mathcal{D}, \epsilon)\beta k} + \min\{k, \dim_{\mathrm{DE}}(\mathcal{G}, \mathcal{D}, \epsilon)\}C + k\omega \right).$$

**Lemma F.2** (Lemma 11 of Jiang et al. (2017)). *Consider a closed and bounded set* $V \subset \mathbb{R}^d$ *and a vector* $p \in \mathbb{R}^d$. *Let* $B$ *be any origin-centered enclosing ellipsoid of* $V$. *Suppose there exists* $v \in V$ *such that* $p^\top v \ge \kappa$ *and define* $B_+$ *as the minimum volume enclosing ellipsoid of* $\{v \in B : |p^\top v| \le \frac{\kappa}{3\sqrt{d}}\}$. *With* $vol(\cdot)$ *denoting the (Lebesgue) volume, we have:*

$$\frac{vol(B_+)}{vol(B)} \le \frac{3}{5}.$$

