# OpenReview forum: "Towards General Function Approximation in Zero-Sum Markov Games"
_ICLR.cc/2022/Conference — ICLR 2022 Poster_

### Official Review · Reviewer_VsQZ · 2021-10-28

**Correctness:** 4
**Technical Novelty And Significance:** 2
**Empirical Novelty And Significance:** Not applicable
**Recommendation:** 6
**Confidence:** 2

**Main Review:**

Strengths:

The presentation is mostly clear and to the point. The main results are related to the relevant literature. A particular strong feature of this paper is a detailed appendix containing the main proofs of the paper. The appendix is mostly well written and one obtains a good overview

Weaknesses:
I See a couple of issues with the references. Some entries seem to be duplicated, or not cited properly. This seems to be the case here:
Julien Perolat, Bruno Scherrer, Bilal Piot, and Olivier Pietquin. Approximate dynamic programming for two-player zero-sum Markov games. In International Conference on Machine Learning, pp. 1321–1329, 2015a.
Julien Perolat, Bruno Scherrer, Bilal Piot, and Olivier Pietquin. Approximate dynamic programming for two-player zero-sum markov games. International Conference on Machine Learning, 37:1321–1329, 2015b.
Ian Osband and Benjamin Van Roy. Model-based reinforcement learning and the eluder dimension. arXiv preprint arXiv:1406.1853, 2014.  (This is a NIPS2014) paper
And at many more places. Note that reference Yulai Zhao, Yuandong Tian, Jason D. Lee, and Simon S. Du. Provably efficient policy gradient methods for two-player zero-sum markov games, 2021.
Is not even properly referenced.

-) I do not understand the „Bellman operator“ defined in eq. (1). Usually, the value is computed outside of the expectation, and not inside. I would also not know what value iteration applied to this operator would deliver. Note that the Bellman operator your introduce is not the one defined in the paper you are citing at this stage: Julien Perolat, Bruno Scherrer, Bilal Piot, and Olivier Pietquin. Approximate dynamic programming for two-player zero-sum markov games. International Conference on Machine Learning, 37:1321–1329, 2015b. In that reference you clearly see that the operator approach of stochastic games is follows, which is the mathematically correct way to define equilibrium in Markov games.

-) Why is v_{\pi}^* or \pi^*_{\nu} unique?

-)The definition of the measures \pi_{f})_{h} is confusing. First, I belief there is a typo on the sentence preceding sentence: It should now be f\in F_{h}, but rather f\in F, doesn’t it? Only then I would understand that you want to construct a sequence of measure-valued policies indexed by the approximating function sequence f and the time horizon.

-) The proposed minimax Eluder dimension looks like a special case of the Bellman Eluder dimension on, previously introduced in the context of MDPs. What is the rational of looking at Dirac measures in this definition? Also, if the state space is not compact, I am wondering if this notion is a suitable concept. What is the minimax Eluder dimension on the real line?

-) Algorithm 1 is a cutting-plane method in the sequence space of reward functions. If I understood it correctly, the round k profile of rewards is chosen as the best reward function at time 1 of the Markov game, when all per-period policies are myopically min-maxed. I am wondering how this is computed. To me the argmax defining $f^{k}$ reads like an optimization problem over a space of functions, and I do not see how this can be done efficiently in the present setting. Note that you do not assume the state space to the finite (unless I missed something). This looks to me like an extremely challenging task and some more explanation would be needed to understand the hardness of this step.

-) After eq. (8) the set \mathcal{Z}_{\rho} has not been defined before I think.


**Summary Of The Paper:**

This paper considers zero-sum Markov games in a general regime where the model is parameterized by general function classes. The goal of this research is to investigate reinforcement learning algorithms that learn a Nash policy in a trial-and-error fashion. The level of generality is achieved by introducing two complexity measures: The minimax Eluder dimension and the witness rank. The main result three algorithms with provable regret upper bounds involving covering numbers and the minimax Eluder dimension.

**Summary Of The Review:**

The paper comes with a very detailed appendix containing all the necessary proofs. This is a big plus of the paper. I am not quite sure, though, how novel the presented results are. At the same time I am admitting that I am not an expert on MDPs or RL. From an algorithmic perspective I have some doubts about the computational complexity of the algorithms presented here, but it seems this is a general problem in this literature, so again I would not put much weight on my critique here.

What I find quite confusing is the use of classical concepts, like the Shapley operator (the analogue of the Bellman operator in stochastic games), is defined in a way that I have not seen before. In the operator approach to stochastic games, it is used to define an extension of dynamic programming to competitive settings. A key property is that the Shapley operator is a contraction, and this drives a value iteration kind of algorithm. I am not sure how these classical results translate in the present formulation, as the operator is different and it is not clear whether contraction properties can be used. The question is then to me how one can relate regret and equilibrium. Note that we know that regret minimization and Nash equilibrium are not the same, and even in zero sum games usually regret minimization implies something about the ergodic average, but not the last iterate. This confuses me quite a bit in this paper (and the cited references, to be fair).

---

> ### Author Response · Authors · 2021-11-22
> **Thank you for your valuable feedback.**
>
> Thank you for your valuable feedback. We address the questions and comments in the following. We hope given these clarifications you will consider increasing your score.
>
>  - "Issues with the references."
>
> We thank the reviewer for pointing out this issue. We have corrected these references in the revised version.
>
> - "The Bellman operator. “I do not understand the „Bellman operator“ defined in eq. (1). Usually, the value is computed outside of the expectation, and not inside. I would also not know what value iteration applied to this operator would deliver.”
>
> The Bellman operators on value functions are given in section 2.2 of Perolat et al. (2015). We use the Bellman operators on action-value functions (Q-functions), which are analogously given in section 4.2 of Perolat et al. (2015). The difference is that Perolat et al. (2015) considers discounted Markov Games (where the rewards are discounted by factor $\gamma$), while our work considers finite horizon Markov Games that do not have discounted rewards. When $\gamma > 0$, it is true that these operators are contractions in $\ell_\infty$ norms, and dynamic programming similarly works in finite horizon undiscounted settings. However these methods require stronger query oracles (such as generative models where the agent can directly estimate the transitions and rewards in any state-action pair), while we consider the online setting where the agents can only start at initial states and rollout the MG.
>
> - "The question is then to me how one can relate regret and equilibrium. "
>
> In the decoupled setting, we compete with an unknown opponent. The regret is measured by the difference between the value of the game (which is the best one can achieve if the opponent always plays the best response) and the cumulative sum of reward mi=0, and therefore regret = 0 implies we learn the value of the game.
>
> In the coordinated setting, we aim to find the approximate Nash equilibrium. The regret is measured by the duality gap and therefore regret = 0 means we learn the NE.
>
> - "Why is v_{\pi}^* or \pi^*_{\nu} unique?"
>
> The value $V^*$ is unique, but $\nu_\pi^*$ and $\pi_\nu^*$ may not be unique. The $\nu_\pi^*$ is defined such that it is a policy that gives the minimum value of the target function $V^{\pi,\nu}$, we do not require it to be unique.
>
> - "The definition of the measures $(\pi_f)_h$ is confusing. First, I belief there is a typo on the sentence preceding sentence: It should now be f\in F{h}, but rather f\in F, doesn’t it?"
>
> Thanks for point out this! You are right that it constructs a sequence of policies $\pi_f = (\pi_f)_1 \times \cdots \times (\pi_f)_H$ indexed by the function sequence $f = f_1 \times \cdots \times f_H$  and the time horizon $h$. We have corrected this in revision.
>
> - "What is the rational of looking at Dirac measures in this definition? Also, if the state space is not compact, I am wondering if this notion is a suitable concept. What is the minimax Eluder dimension on the real line?"
>
> We consider Dirac measures because they are simple examples of candidate policies used in MDP. It is true that in MG people often consider stochastic policies, so we just include them for completeness. The Eluder dimension can be finite with suitable feature parameterizations, such as (compact) linear or kernel function approximations (e.g. Theorem D.1 and discussions that follow). The minimax eluder dimension is infinite when the feature set is unbounded, e.g. when the feature set is the real line. After all, it controls the capacity of function approximation, and learning is in general impossible with unbounded capacity hypothesis sets.
>
> You are right that in general, the function approximation should be compact. However, the state space itself may not be compact, for example with well-posed function approximations such as linear or kernel function approximations (e.g. Theorem D.1 and discussions that follow).
>
> - “Algorithm 1 is a cutting-plane method in the sequence space of reward functions. If I understood it correctly, the round k profile of rewards is chosen as the best reward function at time 1 of the Markov game, when all per-period policies are myopically min-maxed. I am wondering how this is computed.”
>
> We agree that solving the optimization problem is hard. These problems have highly non-convex constraint sets. The proposed methods trade computational efficiency for better sample complexity. Notice that in MDP with bellman complete linear function approximations (Zanette et al. 2020), we match the best known sample complexity upper bound. In this case, however, it is unknown whether the same sample complexity can be achieved with computationally efficient algorithms. We have added discussions in revision.
>
> - "After eq. (8) the set \mathcal{Z}_{\rho} has not been defined before I think."
>
> Thank you for pointing this out! $\mathcal{Z}_{\rho}$ should be $\mathscr{N}(\cal F,{\rho})$, the $\rho$-cover of $\cal F$. We have corrected this typo in revision.

---

> ### Comment · Reviewer_VsQZ · 2021-11-30
> **Post-rebuttal discussion**
>
> The authors made some efforts to improve the submission. However, I am not fully convinced that these efforts really raised the bar. Accordingly I am not going to adjust my grade for this submission.

---

### Official Review · Reviewer_s7Lf · 2021-11-02

**Correctness:** 3
**Technical Novelty And Significance:** 2
**Empirical Novelty And Significance:** Not applicable
**Recommendation:** 3
**Confidence:** 3

**Main Review:**

Strengths:
The paper is well organized and properly presented. Both intuitive design and technical proofs are easy to follow.

Weaknesses:
1. My major concern is about the difference between the model-free algorithms and the work by Jin et al. 2021b. It seems that they share almost the same idea of the algorithm design, similar analysis and results for sampling complexity, especially extending the same measure of Eluder dimension (named minimax Eluder dimension in this paper while multi-agent Eluder dimension in Jin et al. 2021b) without any comparison. Can the auther highlight any technical contribution beyond Jin et al. 2021b?
2. The assumptions of realizability and completeness are not well-motivated. It would be better to provide more convicing evidence that these assumptions are realisitc.
3. One minor question is that whether the sample complexity of the model-based algorithm can be compared with the model-free one?

**Summary Of The Paper:**

This paper studies efficient function approxiamtion in two-player zero-sum Markov games with general function classes. Both decoupled and coordinated settings for learning agents are considered. Model-free algorithms for both settings and model-based algorithm for the leter are provided, all with proved sample complexities.

**Summary Of The Review:**

I think this paper is blow the bar of acceptance.

---

> ### Author Response · Authors · 2021-11-22
> **Thank you for your valuable feedback.**
>
>
> Thank you for your valuable feedback. We address the questions and comments in the following. We hope given these clarifications you will consider increasing your score.
>
> - "My major concern is about the difference between the model-free algorithms and the work by Jin et al. 2021b. It seems that they share almost the same idea of the algorithm design, similar analysis and results for sampling complexity, especially extending the same measure of Eluder dimension (named minimax Eluder dimension in this paper while multi-agent Eluder dimension in Jin et al. 2021b) without any comparison. Can the auther highlight any technical contribution beyond Jin et al. 2021b?"
>
> Jin et al. 2021b is discussed and compared in Appendix A.The discussions make it explicit that Jin et al. 2021b is a concurrent work and is based on different assumptions in the coordinated setting. In comparison, we consider a more general agnostic learning framework and the infinite function class setting. Technically, we maintain a policy-hypothesis confidence set and use a new ‘alternate optimism’ to analyze the regret. These are all different from Jin et al. 2021b.
>
> In addition, we provide model-based methods using bounded witness rank which do not appear in Jin et al. 2021b. This is the first result that model-based function approximations can learn Markov Games with weak assumptions, to the best of our knowledge.
>
> - "The assumptions of realizability and completeness are not well-motivated. It would be better to provide more convicing evidence that these assumptions are realistic."
>
> Realizability and completeness are modeling assumptions. In principle, they are used because approximation error is not something one can prove is small and so in ML we only compare against the best-in-class. Realizability is a necessary assumption in general function approximation, which means the function class contains target action-value function. Completeness says that the function class is closed under the Bellman operator. This assumption is important in *value-based* algorithm utilizing Bellman update. These are all standard assumptions in RL (e.g. Wang et al. 2020, Jin et al. 2020, Jin et al. 2021), and they are all weaker than the assumptions of linear Markov Games in Xie et al. (2020). Further, recent work shows that without completeness there are exponential lower bounds in MDP, meaning that sample efficient learning may *not* be possible without them.
>
> - "One minor question is that whether the sample complexity of the model-based algorithm can be compared with the model-free one?"
>
> In principle, model-based and model-free methods are not directly comparable due to imposing different assumptions. The former consider function approximations to model the environment, which are usually more powerful. Furthermore, it is not clear that minimax Bellman Eluder can subsume the witness rank as a special case (this is even *not* true in MDP), and thus the sample complexity bound in model based setting can *not* be covered by the model-free result.

---

> > ### Comment · Reviewer_s7Lf · 2021-11-23
> > **RE**
> >
> > Thanks for your clarification.

---

> > > ### Author Response · Authors · 2021-11-24
> > > **RE**
> > >
> > > Dear Reviewer, we are wondering if you have any further questions. Although we cannot upload new versions, we are happy to address your additional comments and incorporate your suggestions in the final version.

---

### Official Review · Reviewer_mNjL · 2021-11-02

**Correctness:** 4
**Technical Novelty And Significance:** 2
**Empirical Novelty And Significance:** Not applicable
**Recommendation:** 6
**Confidence:** 3

**Main Review:**

The former works under a (new) notion of Eluder dimension that is appropriate for games, and the algorithm itself is a variant of Jin et al 2021.

The second works under a notion similar to the witnessed rank of Sun et al 2019a, and the algorithm is also inspired by theirs.

I have few remarks.
- The work alludes to general function approximation; I advise caution with the wording, as we do not have yet a full characterization of general function approximation in RL. Besides, the notion of Eluder dimension does not seem to give rise to models much more general than generalized linear models to the best of my understanding.
- The work seems to be a `collage’ of different ideas, namely the Bellman Eluder dimension and witnessed rank. The authors should explain in much better depth why they use one instead of the others in the two different settings.
- Most of the ideas in this work have been introduced previously (in non games)
- The lack of computational tractability should be emphasized much better.


**Summary Of The Paper:**

The paper discussed Markov games with `general’ function approximation. They consider two settings: the decoupled one where the player does not observe the opponent’s policy and the coordinated one where `optimistic planning is performed for both.



**Summary Of The Review:**

While I listed several critical comments to this work, on the whole I liked the work because it still makes a non-trivial contribution to the theory of Markov games.

---

> ### Author Response · Authors · 2021-11-22
> **Thank you for your valuable feedback.**
>
> Thank you for your valuable feedback. We address the questions and comments in the following. We hope given these clarifications you will consider increasing your score.
>
> - “The work alludes to general function approximation; I advise caution with the wording, as we do not have yet a full characterization of general function approximation in RL. Besides, the notion of Eluder dimension does not seem to give rise to models much more general than generalized linear models to the best of my understanding.”
>
> Thanks for your suggestions. In revision, we have changed the wording in the introduction, section 4.2, and the conclusion to avoid the misconceptions that this work fully resolves general function approximations. Note that, however, we do study and make a step towards understanding general function approximations in multi-agent RL. By general function approximation, we mean using an abstract function class to approximate the model or value functions. The function class can be *arbitrary* and does not need to possess linear structures (e.g. value functions have low-rank representations or transition probabilities take the form of linear mixtures). We use minimax eluder dimensions and witness ranks to measure the complexity of function classes. These notions of capacity is general because many function approximations have low minimax eluder dimensions, including linear, generalized linear, RKHS, NTK, and etc.
>
> - “The work seems to be a 'collage’ of different ideas, namely the Bellman Eluder dimension and witnessed rank. The authors should explain in much better depth why they use one instead of the others in the two different settings.”
>
> Bellman eluder dimension and witness rank are the state-of-the-art complexity measures for value and model function approximations in MDP. They can capture a rich set of learnable structures in MDP and enjoy good sample complexity. Bellman eluder dimension does not capture the witness rank. Overall, it is non-trivial to generalize these complexity measures to games, because in Markov Games both agents can influence the state transitions. These lead to complicated concentration issues and distribution shift issues discussed in section 1.2 and section 4.1.1. Addressing these issues requires non-trivial algorithmic ideas.
>
> - “Most of the ideas in this work have been introduced previously (in non games)”
>
> It is challenging to generalize from MDP to MG with more general function approximations than linear or tabular, as the target of minimizing the duality gap is different from maximizing a single objective function. First, we need to address the exploration and exploitation tradeoff in the context of MG, where both players affect the state transitions and rewards. Second, we need a complexity measure to quantify the capacity of the function class. The first is particularly challenging as briefly demonstrated in section 1.2. To give an illustrative example, consider a toy case where both agents alternately exploit each other's policy by optimistically choosing an action-value function from the function class. This can only make sure that $V^{\pi^{f_1}_\nu,\nu} - V^{\pi,\nu^{f_2}_\pi}$ is small for $f_1,f_2$ chosen by P1 and P2 respectively. However this is not the duality gap and thus is meaningless for our target. The second is also difficult because for more general hypothesis classes, it is complicated to determine the width of the confidence set.  Furthermore, it is unclear if optimizing the optimistic objectives will lead to convergence to Nash Equilibrium.
>
> To deal with the first issue, we use ‘alternate optimism’ to construct a confidence set of policy-hypothesis pairs. For the second problem, we develop a series of regret decomposition lemma that decouple the duality gap by Bellman errors in each time step. Coincidentally, these bellman errors can be optimized by the ‘alternate optimism’. These techniques do not appear in previous work, to the best of our knowledge.
>
> - “The lack of computational tractability should be emphasized much better.”
>
> Thanks for this suggestion. We have added discussions in revision.

---

### Official Review · Reviewer_cKpw · 2021-11-06

**Correctness:** 4
**Technical Novelty And Significance:** 3
**Empirical Novelty And Significance:** Not applicable
**Recommendation:** 6
**Confidence:** 3

**Main Review:**

The authors develop algorithms for optimistic nash eliminated
for zero-sum two-player markov games
based on value-function elimination (Zhang 2017).
The contributions are clear in the introduction and the technical challenges section provide good explanation of which concentration bounds and ideas of optimism are employed in the paper.

It would be useful to have some clear definition of what decoupled and coordinated means in this case.
How do those results relate to similar settings in single agent learning?

The discussion feels lacking. I would expect some more commentary comparing the various algorithms. Currently they are treated mostly as separate algorithms. Is there a unifying framework of the three algorithms that you could expand on? Or comment on how the regret bounds compare with each other? When to use one over the other. I think addressing this could make the story more cohesive.

The contributions seem significant for the zero-sum markov games settings.
The motivation for constructing the algorithms in section
3.2, 4.1.1 and 4.2.1 is a bit lacking. What are you trying to do and why are you doing this way? Perhaps some transition sentences could help with clarity.
A minor but related comment: adding line number references in those sections can help make it more clear.

It would be nice to see commentary on the significance of the informal theorems since the proof is omitted. Perhaps comment on whether it is a surprising or expected result.



**Summary Of The Paper:**

This paper presents sample efficient algorithms for learning in two-player zero-sum markov games.
The algorithms are for decoupled and coordinated settings, the latter of which is based on 'alternate optimism'.
The authors also extend the Eluder dimension of MDPs to to zero-sum markov games using the minimax Bellman operator.


**Summary Of The Review:**

The authors present a wide range of results on efficient algorithms for zero-sum markov games.
These results leverage several previous works on value-function elimination using the Bellman factorization and Bellman Eluder Dimension.
The contributions seem to be significant. The correctness seems good but it is possible i did not understand some parts of the submission and unfamiliar with some piece of relate work. The technical novelty seem high, providing new theoretical insights into minimax markov games.

---

> ### Author Response · Authors · 2021-11-22
> **Thank you for your valuable feedback.**
>
> Thank you for your valuable feedback. We address the questions and comments in the following. We hope given these clarifications you will consider increasing your score.
>
> - "It would be useful to have some clear definition of what decoupled and coordinated means in this case. How do those results relate to similar settings in single agent learning?"
>
>  In the decoupled setting, we control one agent to compete with a potentially adversarial opponent. The regret in the decoupled setting reduces to the regret in the single-agent setting when the opponent is a dummy agent. Therefore, the result reduces to the $\sqrt{d_{BE} \log (|{\cal F}|) T}$ result in  MDP (Jiang et al. 2021a).
>
> In the coordinated setting, we control two agents and the target is to find the Nash Equilibrium. Hence the regret is measured by the duality gaps of proposed policy pairs. This result is fundamentally different from the result in MDP because here the objective is to find the Nash Equilibrium, while the objective in MDP is to find the policy that maximizes rewards.
>
> - "The motivation for constructing the algorithms in section 3.2, 4.1.1 and 4.2.1 is a bit lacking. What are you trying to do and why are you doing this way? It would be nice to see commentary on the significance of the informal theorems since the proof is omitted. Perhaps comment on whether it is a surprising or expected result."
>
> All algorithms are based on a similar idea: using optimism in the initial state to address exploration and exploitation tradeoff and constructing a confidence set of hypotheses based on small Bellman errors. The differences lie in how optimism is played and which Bellman error is chosen. We have added more discussions in revision.
>
> In section 3.2, the target is minimizing the difference between the value of MG and the cumulative sum of reward (see Eq.(3)). Therefore the optimism is conducted in a way that assumes the opponent always plays adversarially Line 5. Further, the regret in Eq.(3) decouples into Bellman errors in the form of Eq.(1) (see Lemma B.5), so we use the corresponding Bellman error in confidence construction in Line 11.
>
> In section 4.1.1 and 4.2.1, the target is to minimize the duality gap (see Eq.(4)). Therefore the optimism is different and more complicated to ensure convergence to NE. Alternate optimism in Line 4-5 in AOME alternately exploits the opponents by choosing an optimistic model and assuming opponents playing adversarially. Line 5-6 in AOVE is slightly different but has similar ideas. The Bellman error uses Eq.(2) because we show that the duality gap can be decoupled into the corresponding forms in Lemma C.12.
>
> - "I would expect some more commentary comparing the various algorithms. Currently they are treated mostly as separate algorithms. Is there a unifying framework of the three algorithms that you could expand on? Or comment on how the regret bounds compare with each other?"
>
> Thank you for this advice. Please find our responses in the previous bullet point. We have added more discussions and explained the similarities and differences between algorithms in the revision.

---

### Author Response · Authors · 2021-11-22
**We made revisions based on the feedback of the reviewers.**

We thank all the reviewers for their time and efforts in helping us improve the quality of the paper. We summarize the main changes in revision.

- Discuss computational complexity: We added discussions of the computational complexity of proposed algorithms.
- Fix typos: We fixed the typos in ${\cal Z}_\rho$, the definition of the measures $(\pi_f)_h$, and bugs in reference.
- Add discussions of algorithms: We added discussions of the motivations behind algorithms and how they are related to each other and to the single-agent setting.
- Wording about general function approximation: We polished the wording in several appearances of “general function approximations”.
- Discussions of assumptions: We added discussions on the assumptions of realizability and completeness.

---

### Decision · Program_Chairs · 2022-01-20

**Decision:**

Accept (Poster)

**Comment:**

Summary: The paper discusses Markov games with general function approximation, and investigates in particular reinforcement learning algorithms that learn a Nash policy in a trial-and-error fashion. They consider two settings: the decoupled one where the player does not observe the opponent’s policy and the coordinated one where optimistic planning is performed for both. The main contribution is a new complexity measure called Minimax Eluder dimension which is used to control the regret of the proposed algorithms.

Discussions: The reviewers raised many minor concerns regarding the writing (typos of missing notation) and the clarity (missing discussions and explanations), which were addressed during the discussion phase. In light of this revision, the committee and myself judge that the paper should be accepted to ICLR.

Decision: Accept